

# The contribution of land-use change versus climate variability to the 1940s CO$_2$ plateau: Former Soviet Union as a test case

Ana Bastos[1], Anna Peregon[1], Érico A. Gani[2], Sergey Khudyaev[3], Chao Yue[1], Wei Li[1], Célia Gouveia[2,4], and Philippe Ciais[1]

[1]Laboratoire des Sciences du Climat et de l'Environnement, LSCE/IPSL, CEA-CNRS-UVSQ, Université Paris-Saclay, 91191 Gif-sur-Yvette, France
[2]Instituto Dom Luiz, Universidade de Lisboa, Portugal
[3]Institute of Soil Science and Agrochemistry, Siberian Branch Russian Academy of Sciences (SB RAS), Novosibirsk, 630090, Pr. Akademika Lavrentyeva, 8/2, Russia
[4]Instituto Português do mar e da atmosfera (IPMA), Lisboa, Portugal

*Correspondence to:* Ana Bastos (ana.bastos@lsce.ipsl.fr)

**Abstract.** According to the ice-core record, atmospheric CO$_2$ growth rate (*plateau*) stalled during the 1940s, in spite of maintained anthropogenic emissions from fossil fuel burning and land-use change. Bastos et al. (2016) have shown that the state-of-the-art reconstructions of CO$_2$ sources and sinks do not allow closing the global CO$_2$ budget during this period. Their study indicates that even considering an enhancement of the ocean sink, still a gap sink of 0.4-1.5PgC.yr$^{-1}$ in terrestrial ecosystems

is needed to explain the CO$_2$ stabilization. They hypothesised that (i) the major socioeconomic and demographic disruptions during World War II (WWII) may have led to massive land-abandonment, resulting in an additional sink from regrowing natural vegetation which is not accounted for in most reconstructions and/or (ii) the warming registered at the same time, especially in the high-latitudes, might have led to increased vegetation growth and an enhancement of the natural sink.

Here, we test the different contributions of these two factors in the Former Soviet Union (FSU), motivated by several reasons.
On the one hand, the territory of the FSU encompasses 15% of the terrestrial surface, 20% of the global soil organic carbon pool and is responsible for a considerable fraction of the present-day terrestrial CO$_2$ sink. On the other hand, heavy economic and demographic losses have been registered in FSU during WWII, together with likely decrease in farmland due to occupation, destruction of infrastructure and shortages of manpower.

Here we present a newly compiled dataset of annual agricultural area in FSU, which better matches other socioeconomic
indicators and reports a decrease in cropland of ca. 62Mha between 1940-1943. We use an updated version of the land-surface model ORCHIDEE, ORCHIDEE-MICT, which is specifically developed to better represent high-latitude processes to simulate the carbon fluxes in terrestrial ecosystems over the 20$^{th}$ century. Using our new cropland dataset, we test the different contributions of the land-use change and the decadal warming reported in the 1940s to explain the *plateau*. As reference, we compare our results with the gap sink estimated by the group of land-surface models in Bastos et al. (2016): 0.7PgC.yr$^{-1}$.

We find that the massive cropland decrease between 1940-1943, even if short-termed, could result in an additional decadal sink of 0.04-0.07PgC.yr$^{-1}$, i.e. 6-10% of the gap sink required to explain the *plateau*. The ORCHIDEE-MICT simulations also indicate a very strong enhancement of the terrestrial sink by 0.4PgC.yr$^{-1}$, explaining about 60% of the gap sink from the



TRENDYv4 models. This enhancement is mainly explained by tree-growth in high-latitudes coincident with strongest warming sustained over the 1940-1949 decade, which is not captured by any of the other land-surface models.

Even if land-abandonment during WWII might contribute to a relatively small fraction of the sink required to explain the *plateau*, it is still non-negligible, especially since such events have likely been registered in other regions. The vegetation growth in high-latitudes simulated by ORCHIDEE-MICT and absent in other models appears to be supported by tree-ring records, highlighting the relevance of improving the representation of high-latitude hydrological and soil processes in order to better capture decadal variability in the terrestrial $CO_2$ sink.

## 1 Introduction

During the mid-20[th] century, the ice-core record indicates that atmospheric $CO_2$ appears to have stabilized (Rubino et al., 2013), in spite of continued fossil fuel (Boden et al., 2009) and $CO_2$ emissions from land-use change (LUC) (Brovkin et al., 2004; Houghton et al., 2012). Several hypotheses have been discussed, with earlier works suggesting a strong enhancement of the ocean sink (Joos et al., 1999; Rubino et al., 2013), but others pointing to an increased terrestrial sink (Rafelski et al., 2009). In fact, the reasons leading to this one-time stabilization of $CO_2$ during the anthropocene are still not fully clear, as the current reconstructions of $CO_2$ sources and sinks over the 20[th] century do not allow closing the $CO_2$ budget during this period (Bastos et al., 2016).

In their study, Bastos et al. (2016) show that the current sink gap, defined as the sink not accounted in current land and ocean models compared to the sink needed to explain observations during the 1940s is of 0.9-2.0PgC.yr$^{-1}$. They suggest that variability in the ocean sink alone is not likely to explain more than about 0.5PgC.yr$^{-1}$, implying an additional terrestrial sink gap of 0.4-1.5PgC.yr$^{-1}$. They showed that the main uncertainty in land carbon models' estimates of the sink gap derives from the very large spread in estimates of land-use change emissions, which in the first half of the 20[th] century contributed approximately as much as fossil fuel burning to the total anthropogenic emissions. Given the timing of the *plateau* (ca. 1940 until 1950), Bastos et al. (2016) suggest that the stall in atmospheric $CO_2$ may be related with widespread abandonment of croplands due to socioeconomic and demographic collapses imposed by the second World War (WWII) in some countries, currently not accounted for in land-use area change datasets.

All model studies applying land-use reconstructions from Klein Goldewijk et al. (2011) (HYDE 3.1) and Hurtt et al. (2011) (LUH1), itself based on HYDE 3.1, are based on FAO data for agricultural areas only after 1961. For earlier periods, they rely on extrapolating country-level agricultural area backwards using total population dynamics, and cropland area per-capita is only allowed to vary very little. This assumption may be unrealistic for certain periods in history. Furthermore, these data have only decadal resolution (annual values are calculated by interpolating between two decadal values), as they are intended to capture longer-term changes. However, in order to understand the processes responsible for the 1940s plateau, higher temporal detail is required in order to fully capture the drastic changes imposed by war conditions, in particular for countries that suffered large population displacements and casualties and destruction of agricultural infrastructure.



Based on model experiments between 800 and 1850 (AD), Pongratz et al. (2011) suggest that massive land-abandonement due to wars may indeed induce an increased terrestrial sink. However, the authors also point that for events of short duration, terrestrial fluxes remain dominated by delayed emissions from past LUC, since regrowth is a slower process. Relying on model simulations, Brovkin et al. (2004) also concluded that LUC was unlikely the sole responsible for the stalling of atmospheric

$CO_2$ during the 1940s. Nevertheless, as Bastos et al. (2016) pointed, such processes could contribute at least to a fraction of the 0.4-1.5PgC.yr$^{-1}$ additional sink required to explain the *plateau*.

In this context, the Former Soviet Union (FSU) is a region of particular interest. The total territory of FSU represents almost 15% of the terrestrial surface, accounting for a significant fraction of the global terrestrial sink (0.8-1.0 in early 1990s (Kudeyarov, 2000)), and soil organic carbon pools (20% of the global pool (Kurganova et al., 2010)). Furthermore, the society

in FSU during the first half of the 20$^{th}$ century was highly rural, with only 18% of the population living in cities in 1926, and 33% in 1939 (Nove, 1982). In 1927-28, output from agriculture contributed 42% of gross industrial and agricultural production combined, a number that was reduced over the following years due to the massive industrialization effort. The war was responsible for the death of 26.6 million people (14% of the population) (Harrison, 2000b), and furthermore, at least 10 million people are estimated to have been evacuated from the western front, where most agricultural area was located (Linz,

1984). During the war years, while industrial production decreased only by 8% between 1940 and 1945, agricultural production recessed by 40% (Nove, 1982). The decrease in grain production during the peak of the war was even more extreme: 50–69% lower in 1943 than in 1940 (Nove, 1982; Sapir, 1989).

Statistics about agricultural areas in the Soviet Union during 1940–1945 are scarce and contradictory. LUH1 report a 6.6 Mha decrease in crop area between 1940 and 1950 while Lyuri et al. (2010) point to a decrease in in crop area within the

territory of the Russian Federation of ca. 25 Mha for the same period. In fact, land-abandonement was possibly even higher in Ukraine and Belarus, due to shortage of manpower (mortality, evacuation and war mobilization (Nove, 1982)) and destruction of infrastructure. Even though a massive farmland re-location plan was set in motion during the war to move agriculture away from the war from (Linz, 1984), the new farmland created likely did not compensate land abandonement in the affected war territories. Nove (1982) reports a huge drop in sown area of 43.1 Mha between 1940 and 1942. Cropland is reported to have

recovered slowly, returning to pre-war levels only in the early 1950s (Lyuri et al., 2010; Nove, 1982).

Since the different studies report about different periods and regions, the numbers are hard to reconcile. Nevertheless, it is clear that the figures provided by Klein Goldewijk et al. (2011) and Hurtt et al. (2011) – whose data were used to perform the land-surface model simulations and bookkeeping inventories described in Bastos et al. (2016) – likely overlooked these drastic, but short-termed land use changes.

Here we present a newly compiled annual dataset of agricultural area in FSU based on Russian Empire (during the pre-Soviet period, 1913–1916) and Soviet statistics between 1917 and 1961. We then derive new land-cover maps based on the one from LUH1 and our data, and compare the impact of different LUC trajectories on the land carbon sink in FSU. We use the most recent version of the land-surface model ORCHIDEE-MICT that has an improved representation of high-latitude hydrology, soil carbon dynamics and phenology and is, therefore, better tailored for the study area. We compare the $CO_2$ fluxes simulated

by ORCHIDEE-MICT with the ones estimated by the TRENDYv4 land-surface models used in Bastos et al. (Bastos et al.,





2016; Sitch et al., 2015) in order to evaluate the likely contribution of the fluctuations in cropland area in FSU from our new land use dataset during the 1940s to the global $CO_2$ sink. Finally, since ORCHIDEE-MICT should also improve the response of high-latitude vegetation to climate variations, we assess the relevance of the warming in the 1940s reported in Bastos et al. (Bastos et al., 2016) to compare the relative contribution of LUC and climate to the FSU sink during the *plateau* period.

## 2 Data

### 2.1 Land Use / Land Cover

#### 2.1.1 Russian and Soviet crop area

The new dataset of FSU agriculture area comprises the data collected from official national statistics from Lyuri et al. (2010). National statistics were provided for the Former Soviet Union (FSU) during the period 1917-1961, and the Russian Empire during the pre-Soviet period starting from 1913 (dataset and full list of references in Supplementary Data). The data for Russian Empire is presented in its actual borders, i.e. not conform to FSU borders, and originally derived in obsolete Russian units of land area (dessiatina, 1 Des=1.09 ha). At the time of the Soviet Union, the statistics provided total cropland area (Mio ha), and cropland area in each of the 15 Federal Republics (thousands ha). Even though the borders of the Russian Empire and FSU were different, both covered the areas which account for most of the cropland area (Russia, Ukraine, Kazakhstan, see Figure A2) and therefore we aggregate both datasets.

The total agricultural area is divided into regional values when available and includes winter and spring crops, industrial crops and sown area for fodder. We placed special focus on collecting data for the area of cropland land on occupied and non-occupied territories made in a subset of the data during the World War II (1941-1945, Supplementary Data). Total cropland area for the FSU aggregated for all types of agriculture products (hereafter referred to simply as FSU–NEW) from these sources is shown in Figure 1. As an additional source of information, we use the values of sown area reported by Nove (1982), which are provided as % relative to 1940 (Figure A2). However, it is worth noting that these data are not necessarily independent from ours, as it likely relies on some of the sources used here.

#### 2.1.2 Agricultural area from reference datasets

We use for comparison with our new dataset the land-use harmonization from Hurtt et al. (2011) (LUH1) since it is the one used to force the dynamic global vegetation models in Bastos et al. (Bastos et al., 2016). The LUH1 provides annual values at half-degree from 1500 until 2100 of fractional data on cropland, pasture, primary vegetation, and secondary vegetation, as well as the underlying transitions between land-use states. In this dataset cropland, pasture, urban, and ice/water fractions between 1500 and 2005 are calculated based on the HYDE 3.1 database (Klein Goldewijk et al., 2011) that provides gridded time series of historical population and land-use data for the Holocene. The HYDE 3.1 database relies on U.N. Food and Agricultural Organisation data (FAO, 2008) national statistics of agricultural areas from 1961 onwards, and extrapolates these country-level estimates backwards in time using population dynamics, with cropland and pasture values per capita allowed to change only





slightly prior to 1961(Klein Goldewijk et al., 2011). The LUH dataset further includes wood harvest and shifting cultivation. This data was used to produce the land-cover maps to force the land-surface model, with the forest, grassland and crop classes being converted to the model's plant functional types following the method by Poulter et al. (2011). In the model, grazing and pasture management are not explicitly represented and pastures are therefore treated as natural grassland. Henceforth, we

refer to this dataset simply as FSU–REF. It should be noted that some differences between the original LUH/HYDE and the FSU–REF are expected due to the PFT conversion. The corresponding area occupied by crop PFTs in FSU–REF is shown in Figure 1.

## 2.2 Socio-economical statistics

In order to assess how the datasets described in Section 2.1 capture the transitions that occurred in FSU during the first

decades of the 20[th] century, we compare them with several sources of data on population and gross domestic product (GDP). The relationship between population and economic output with total crop area likely changed over the 20[th] century due to agricultural mechanization and fertilization or to rural exodus. Nevertheless, they provide reasonable proxies to evaluate the variability of crop area reconstructed by FSU–REF and the FSU–NEW statistics.

Information on total population in the FSU was collected from national statistics (Supplementary Data). Total population was

partitioned as urban and rural population, however, urban and rural population are defined from the designation of permanent residence, which does not necessarily mean population working in agriculture. We, therefore, collected information on active population and population working in agriculture. Even though less data is available (especially for the latter), both variables present similar variations during the WWII period.

The Maddison Project (Bolt and van Zanden, 2014) extends the original work of Maddison (2001) in order to provide per-

country population and GDP data extending back (when available) to Roman times. Data are provided for the FSU from 1928 onwards, although during 1941-1945 GDP values are constant, which is possibly due to lack of data. Harrison (2000a) focused on the Soviet economy during the war period and provides estimates of GDP from 1940 to 1945 that was used to fill in the information lacking in Maddison's data. The socioeconomic indicators described above are shown, together with the crop area estimates in FSU–REF and FSU–NEW in Figure 1. We further collected information about total grain production in FSU from

three different sources based on national statistics (Nove, 1982; Sapir, 1989; Davies et al., 1994) in order to evaluate the model simulation results. These statistics were based on grain harvest reported farmers but, at least between 1920 and 1940, these numbers were then revised by state statisticians that applied "correction factors" based on sown area that likely exaggerated the production (Davies et al., 1994). In their original works, Nove (1982), Sapir (1989) and Davies et al. (1994) have tried to obtain values closer to the original raw data using different assumptions, therefore we keep them as separate evaluation datasets of

crop grain harvest in ORCHIDEE-MICT simulations.

## 2.3 ORCHIDEE-MICT

ORCHIDEE is a global land-surface model (Krinner et al., 2005) representing the main energy, hydrological and carbon cycling processes in land ecosystems. Here we use the updated version of the land surface model ORCHIDEE which is specifically



developed for high-latitude processes: ORCHIDEE-MICT (Zhu et al., 2015; Guimberteau et al., 2017). The model includes an enhanced description of high latitude land surface processes such as an enhanced hydrological balance, the effect of snowpack insulation in winter and its coupling with soil temperature, and an improved description of soil carbon interactions between soil freeze, soil water holding capacity and thermic conductivity. The new soil carbon module was shown to reproduce the

amount of soil carbon in the high latitudes and the seasonal exchange of $CO_2$ resulting from the seasonal imbalance between gross primary productivity (GPP) and total ecosystem respiration (TER). Fire occurrence is simulated using the SPITFIRE fire model as described in Yue et al. (2014), which is well calibrated to simulate boreal fires.

The new data we compiled provides only information about net cropland area changes and hence do not include any information on processes that could result in large differences in the $CO_2$ balance between net/gross LUC, and *ad-hoc* assumptions

would need to be made in order to produce maps with sub-pixel LUC. Therefore, only $CO_2$ fluxes from net land-use change are calculated here, which are based on the difference in the land-use/land-cover class fraction between two years, as described in (Piao et al., 2009).

For crop plant functional types (PFT), a fraction of crop NPP is harvested and consumed directly (Piao et al., 2009). This value likely was not been constant over the whole century, as agricultural harvest practices have undergone drastic changes,

however, in this study, it was set to 0.85 (Hicke and Lobell, 2004). In ORCHIDEE-MICT(Zhu et al., 2015; Guimberteau et al., 2017), crop PFTs parameters are adjusted based on recent observations and are therefore representative of present-day productive crop varieties. Since we are mainly interested in the earlier half of the 20[th] century, we adjusted the crop the maximum rate of carboxylation ($Vc_{max}$) for lower values, calibrated against observations from several cropland sites (Table A1).

## 3   Methods

### 3.1   Updated gridded LUC data

Based on FSU–NEW data for total crop area in FSU (subsection 2.1.1, Figure 1), we produced new spatially explicit maps used to force ORCHIDEE-MICT by combining patterns from FSU–REF and statistics from FSU over administrative units. First we fill the two missing years (1918 and 1919) using a linear adjustment. Since FSU–NEW data always estimate lower cropland area than FSU–REF, in each pixel the crop area was reduced proportionally to the corresponding contribution to the overall FSU crop area (i.e. more crop area is reduced in pixels with high cropland area in FSU–REF) at each time-step to match the FSU statistics in each administrative unit. The country-level cropland area data from FSU–NEW was then compared with the crop area represented in FSU–REF and our updated maps (Figure A3).

Given that we have no additional information about forest or grassland changes in FSU during this period (apart from FSU–

REF), we define two scenarios for the natural vegetation replacing crop area after abandonment. In the first one, crop area in each pixel is replaced by forest cover if forest is already present otherwise it is replaced by grassland (FOR). The second scenario (GRA) is similar, but with grasslands replacing abandoned cropland, if grassland is already present, and otherwise allocated to forests. It should be noted that these two cases correspond to the two extremes of the possible range of forest vs.





grassland trajectories in regions where agricultural area was abandoned. The resulting total forest and grassland areas over
FSU corresponding to each case are shown in Figures A2b, c.

## 3.2 Model simulations

The information about forest and crop and grassland fractional cover and transitions from LUH1 was converted to 2x2 degree
5   lat/lon maps of the 13 PFTs in ORCHIDEE-MICT consisting of bare soil, 8 forest PFTs, 2 crop PFTs (C3 and C4 crops) and
grass PFTs (C3 and C4 grass). We consider only the region corresponding to the FSU, as highlighted in the shaded areas in
Figure A1. The average PFT distribution over the 20[th] century in the FSU region, is shown in Table A1.

    Five factorial simulations were performed in order to test the different contributions of land-use and climate to changes in
terrestrial carbon stocks in FSU during WWII, as summarized in Table 1. Even though we are mainly interested in the period
1940–1950 we extend the simulations to 1999 in order to evaluate the results against observation-based data.

    In order to ensure comparability, we use the same protocol to the land-surface model simulations than for the TRENDY
models (Sitch et al., 2015), analyzed in Bastos et al. (2016). The spinup is performed with cyclic climate (10yr cycle from
1901-1910), land-cover map from FSU–REF in 1860 and constant $CO_2$. All simulations are then started in 1860, forced with
historical observed $CO_2$ concentration between 1860-1999, cyclic climate until 1900 and observation-based historical climate
afterwards (CRU/NCEP v5.4).

    In the baseline simulation ($S_{Ref}$), ORCHIDEE-MICT is forced with historical climate between 1901 and 1999 and prescribed
land-use changes (LUC) from FSU–REF. We perform three other simulations corresponding to different LUC scenarios, one
with constant land-cover map (1860 map from FSU–REF) i.e. no land-use ($S_{noLUC}$), and other two simulations where vegetation
is prescribed from our updated LUC maps, and corresponding to the FOR/GRA scenarios ($S_{FOR}$ and $S_{GRA}$, respectively). All
three use the same climate forcing as $S_{Ref}$. In order to test the influence of climate anomalies in the observed $CO_2$ fluxes, a
simulation was performed in which climate is constant, with the forcing between 1901-1910 being repeated ($S_{Clim}$), in order to
include some degree of interannual variability.

## 4 Results

### 4.1 Agricultural area in FSU

Figure 1 compares cropland area from FSU–REF and FSU–NEW with different socioeconomic indicators. FSU–REF (bold
line, black) estimates systematically higher cropland area than FSU–NEW (black line with + markers), but also much less
variable values. This is consistent with the decadal temporal resolution of the HYDE dataset and with the fact that crop area
before 1961 was reconstructed based on total population (Klein Goldewijk et al., 2011), which also shows smooth variations
(blue line with + markers).

On the contrary, active population (blue circles) presents sharper variations that follow closely the changes in GDP between
1928 and 1970 (cyan line). Even though few data of active population are available during the full period, annual data between





1940-1945 are complete. These values show an increase during the 1930s, peaking in 1940 followed by a sharp drop of about 30% in active population numbers, which is consistent with the values of change in GDP from (Harrison, 2000a). After 1950, GDP and active population increase at the same pace.

The FSU–NEW data (thin black line with + markers) give a decrease in cropland area during the 1910-1920 period, coinciding with the period of Russian participation in Wold War I (1914-1916) and the Soviet revolution (data missing for 1918 and 1919) concurrent with lower numbers of active popultion. Agricultural area subsequently increased until 1940, along with GDP and active population numbers, except the period of famine in 1921-1922. The evolution of cropland area matches closely the variations in GDP (and active population) between 1940 and until 1955, which is consistent with the important role of cropland production in the Soviet economy during this period (Nove, 1982).

The sharp drop in total cropland area between 1940 and 1942 corresponds mainly to a massive cropland abandonement in occupied regions (that dropped from ca. 71Mha in 1940 to 23Mha (68%) in 1943, Supplementary Data). Such decrease is reported in other studies (Nove, 1982; Linz, 1984) to be a result of land abandonmnent, as well as the dislocation of farmland away from the war front during this period.

Even though crop area is updated proportionally to the contribution of each pixel to the FSU total (i.e. does not account explicitly for country level data), most crop areas abandoned are in the western regions (Figure A1). The resulting updated cropland area for each country therefore results in a better match to FSU–NEW data than FSU–REF, especially in the FSU regions contributing more for the total farmland (Figure A3).

## 4.2 Carbon fluxes during the 20th century

In Figure 2a, we compare the net terrestrial uptake (NBP) from $S_{Ref}$ and $S_{noLUC}$ with the corresponding simulations from the TRENDYv4 models (Sitch et al., 2015), that were analyzed for the plateau period in Bastos et al. (2016). Simulated $CO_2$ uptake is generally within the TRENDY inter-model range, except the periods between 1920 and mid-1930s and from ca. 1940 until the mid-1950s, when ORCHIDEE-MICT simulates stronger terrestrial uptake in both simulations. These periods are characterised by strong departures of NBP between $S_{Ref}$ and $S_{Clim}$ (with constant climate, shown in Figure A4), indicating a strong contribution of climate-related anomalies in vegetation growth due to decadal climate variability in addition to LUC, which were not captured by the TRENDY models. These discrepancies during the 1940s will be analyzed in detail further on.

The flux from LUC calculated using FSU–REF map ($S_{Ref}$) remain within the range of TRENDY models during the 20[th] century (Figure 2b). While the TRENDY inter-model average indicates an increasing terrestrial sink due to LUC over the century (negative trend) in FSU, starting from a strong initial source (ca. 0.3PgC.yr$^{-1}$ in early 1900s), ORCHIDEE-MICT simulates a small but relatively stable LUC source. A strong decrease in land use emissions (ELUC obtained by forming the difference between $S_{Ref}$ and $S_{noLUC}$) is simulated by ORCHIDEE-MICT between 1901 and 1915, contemporary with grassland expansion (Figure A2). From 1920 until the mid 1950s, $E_{LUC}$ from $S_{Ref}$ remain close to the TRENDY inter-model mean.

In order to evaluate whether the different simulations produce realistic carbon fluxes and stocks in the recent period, we compare simulated values with observation- based data from previous studies (Table 2). Most of these values are available only for the 1990s and cover different regions of FSU or certain vegetation types. The simulations of ORCHIDEE-MICT show good



agreement with observation-based data of carbon fluxes and pools for the late 20$^{th}$ century (Table 2). Both NPP and NBP are underestimated by 5-10%, but simulations forced with the new vegetation cover maps based on FSU–NEW data result in better agreement of both variables with reference values in the early 1990s (Kurganova et al., 2010; Kudeyarov, 2000). Soil carbon in cropland areas is also close to the literature values (Smith et al., 2007), with the two simulations using FSU–NEW data showing

better agreement. Even if they are forced with the same extent of cropland area, these $S_{FOR}$ and $S_{GRA}$ differ in their estimates of soil C in crops by 1.3 PgC over the FSU, which highlights the relevance of the slow processes and memlegacy effects in carbon fluxes from LUC. Simulated forest C in Russia in 1993 is overestimated by about 4–7% relative to the values reported by Shvidenko and Nilsson (2003) and within their uncertainty range. Total vegetation carbon in the FSU region between 1993 and 1999 is very close to the reported values of Liu et al. (2015) for $S_{Ref}$ and $S_{GRA}$, and overestimated by 13% in $S_{FOR}$.

Generally, the ORCHIDEE-MICT simulations forced by new LUC maps with lower cropland area estimate lower LUC emissions and, in some cases even a LUC-related sink, as during period between 1913 and the early 1920s (coincident with decreasing cropland area during the WWI and the Russian Civil War, but also with the transition between the two datasets) and also during the early 1940s (contemporary with the peak of land abandonment during WWII). The two simulations with FSU–NEW cropland area ($S_{FOR}$ and $_{GRA}$) estimate very high C accumulation in soils starting in 1920 and peaking in the first

years of the 1940s (Figure A5), likely due to the forest area expansion between 1910 and 1920 (also in $S_{Ref}$) combined with the effect of crop area decrease between 1913-1920 due to the correction in FSU–NEW dataset.

Simulated crop productivity (converted to grain production) in FSU from $S_{Ref}$ (Figure 3) is in line with estimates of Soviet grain production until 1945, and captures the increase from the mid 1950s until 1980 reported in Nove (1982), Sapir (1989) and Davies et al. (1994). The simulations using lower cropland extent ($S_{FOR}$,$S_{GRA}$) generally lead to underestimates of grain

harvest, although in some years their estimates are closer to reported values (e.g. around 1930 and between 1940-1945).

### 4.3 Terrestrial uptake in FSU during the 1940s

The strong sink enhancement of the land sink in FSU in the 1940s (about 0.4PgC.yr$^{-1}$ higher than the previous decade) is found in all simulations with historical climate forcing (Figure 2) and is linked with stronger gross primary production (GPP) and leaf-area increase (Figure A4). In spite of a concurrent enhancement of respiration, increased vegetation growth results in

fast C accumulation in biomass and soils (Figure A5). An increase in biomass C is observed in the 1920-1930 (+0.1PgC.yr$^{-1}$), although the one in the 1940s is stronger (+0.4PgC.yr$^{-1}$) and sustained for a longer period. Both periods are preceded by peaks of fire activity, so the enhancement of the natural C sink may be partly due to biomass recovery following fire but, since these variations are inexistent in $S_{Clim}$, also to due to vegetation response to climate variations.

Emissions from LUC in the 1940s using FSU–REF maps are 0.09PgC.yr$^{-1}$, very close to the average E$_{LUC}$ reported by the

TRENDY models (Figure 4). Our two scenarios using lower crop area extent and accounting for abandonment during the 1920s and 1940s lead to significantly lower E$_{LUC}$, 0.05$^{-1}$ and 0.02$^{-1}$ for FOR and GRA scenarios (i.e. 55% and 22% of E$_{LUC-Ref}$) respectively. Such strong differences are not observed in the previous decade (less than 7%) and, therefore, result mainly from the sharp decrease in cropland area observed in FSU–NEW data between 1940 and 1942 which is not captured in FSU–REF as it captures mainly decadal variations.





All simulations forced by historical climate show a strong increase in $CO_2$ uptake starting in the last years of the 1930s, peaking ca. 1945 and persisting until the mid-1950s. This is not represented in any of the TRENDY models (Figure 2) and it is a specific feature of the high latitude carbon cycle representation in ORCHIDEE-MICT. In the 1940s, the inter-model average of NBP from TRENDY simulation S3 (comparable to $S_{Ref}$) is of only $0.04$PgC.yr$^{-1}$, against $0.63$PgC.yr$^{-1}$ for $S_{Ref}$.
The simulation with cyclic climate variability ($S_{Clim}$) also does not estimate such a strong enhancement of NBP. This suggests the occurrence of a decade-long climate anomaly (or a combination of anomalies) leading to increased $CO_2$ uptake of about $0.5$PgC.yr$^{-1}$ ($S_{Ref}$-$S_{Clim}$) in the FSU territory.

As shown in Figure 5 the period 1940-1949 was considerably warmer than the previous decade over most of FSU, and especially in the latitude band between 45-65$^o$N, where temperature anomalies (over the decade) reached values over $0.5^o$C above
the mean value registered in the previous decade. The ORCHIDEE-MICT simulations indicate that the warming observed in this latitude band in the 1940s ($S_{Ref}$) is associated with a large NBP increase sustained during the 1940s of 13gC.m$^{-2}$.yr$^{-1}$ above the previous decade's mean, while the simulation with cyclic climate and variable $CO_2$ concentration only indicates a negligible difference between the two decades (1gC.m$^{-2}$.yr$^{-1}$).

This NBP enhancement in $S_{Ref}$ during the 1940s is associated with a considerable increase in the carbon stocks in the
vegetation in most regions, but especially in the latitudes registering higher temperature anomalies. Changes in vegetation C simulated by $S_{Clim}$ are smaller than in$S_{Ref}$ (and even negative in latitudes above 60$^o$N), which indicates increased growth in response to the 1940s warming in the ORCHIDEE-MICT model. The comparison of soil C changes estimated by $S_{Ref}$ and $S_{Clim}$ indicates lower soil C accumulation due to the high temperature effect on respiration. However, $S_{Ref}$ still estimates an increase in the soil carbon pool during the 1940s, likely because of the even stronger increase in GPP, and soil C increase in the latitudes
above 60$^o$N. Overall, $S_{Ref}$ estimates an average C accumulation in vegetation and soils of $0.12$ kgC.m$^{-2}$.yr$^{-1}$.

Even though the TRENDY models also simulate a small increase of NBP coinciding with the warming in latitudes below 65$^o$N, this increase is only about 30% of the one estimated by ORCHIDEE-MICT. Furthermore, the models diverge in the response of the vegetation and C stocks to the 1940s warming, with most models estimating a small accumulation of C in vegetation, and an overall decrease in the soil, especially where higher temperatures were registered.

## 5   Discussion

In the early decades of the 20$^{th}$ century, the FSU regions were mainly composed by agrarian societies, with a large majority of the population living in rural areas (above 80%, Supplementary Data). At the time, agricultural and industrial production contributed in comparable amount to the national income (13.1 and 18.3 million roubles in 1927, respectively (Nove, 1982)).

Relative variations in population working in agriculture mostly follow variations in total active population, especially in the
beginning of the century. Urban population increased rapidly during the 1930s and after WWII, but only in 1961 did the urban population outweigth rural population. Due to the important role of agriculture in the FSU economy in the early 20$^{th}$ century, a strong relationship between GDP, agricultural area and output is expected. With the expansion of the industry, and along



with the mechanization of agriculture in later decades, the relationship between agricultural area, crop production and GDP is expected to weaken.

Indeed, we find a good agreement in the evolution of cropland area in our new dataset, with active population and GDP values until ca. 1955 (Figure 1), which suggests that our data may better reflect crop area variations during this period. This

implies that using total population to extrapolate country-level crop area (as used in Klein Goldewijk et al. (2011)) may lead to inconsistences in reconstructed LUC, since it does not reflect the effects of migratory fluxes between rural and urban areas (which are highly relevant in the 20[th] century), or variations in active work force due to war (or other disrupting events), or the mobilization of labour to other sectors of the economy (Linz, 1984). Nevertheless, it should be noted that using total population data may still be the best variable to harmonize country-level data at global scale, in a consistent way (e.g. independent of

definitions of "employment"), for earlier periods in history (when other indicators may not be available).

The method used here to update LUC maps does not account explicitly regional dynamics, such as the displacement of farmland from the front and occupied regions during WWII to the eastern countries (Linz, 1984). We deliberately did not account for this displacement, as it would imply changing also natural vegetation fractions in other regions of the FSU (e.g. require forest/grassland removal), and increase the possible inconsistencies between the datasets. Even though such differences

may be relevant at local scale, they are unlikely to significantly change our results for the aggregated FSU. Our data still provides a better match to country-level estimates of crop area (Figure A3) than previous reconstructions.

ORCHIDEE-MICT simulations show the characteristic slow increasing trend of terrestrial uptake over the 20[th] century, consistent with the $CO_2$ fertilisation and warming effect in vegetation productivity (Figures 2 and A4a). Overall, the simulations using the maps derived from FSU–NEW cropland data provide a better agreement with observed carbon fluxes and stocks in

the late 20[th] century, especially $S_{GRA}$. Total soil carbon stocks are overestimated in the recent period by 18–20% (Table 2) relative to the dataset presented in Guimberteau et al. (2017) which are based on observation-based soil organic carbon maps from NCSCD (Hugelius et al., 2013) and HWSD (FAO et al., 2012). However, the simulated soil carbon in crop areas in the late 20[th] century is consistent with the data from Smith et al. (2007), especially for the two simulations with new crop area ($S_{FOR}$ and $S_{GRA}$).

Simulated grain production in FSU is in line with estimates of Soviet grain production until 1945, and captures partially the increase from the mid 1950s until 1980 reported in different works ((Nove, 1982; Sapir, 1989; Davies et al., 1994)). In the simulations performed here, agricultural harvest corresponds to a simple fraction of aboveground NPP in agricultural PFTs and does not consider changes in agricultural practices, such as fertilization inputs or variations in harvest rates due to demographic changes. Therefore it is expected that the grain harvest simulated by ORCHIDEE-MICT should be underestimated for the

second half of the 20[th] century (as shown in Figure 3). It should be noted that the grain production in Soviet reports during the 1920s and 1930s are likely exaggerated due to the use of correction factors for grain production (Davies et al., 1994). Davies et al. (1994) proposes lower values that are usually between the ones using FSU–REF and FSU–NEW cropland areas.

Emissions from land-use change are generally within the range of values estimated by the TRENDY models (Figure 2b), even in the simulations using lower crop area extent. Significant differences between $S_{Ref}$ and $S_{FOR}$, $S_{GRA}$ occur only in two

periods: the one from 1913 to ca. 1925 and in the 1940s. Both correspond to sharp abandonment of cropland in FSU-AGR data





which results in reduced $E_{LUC}$. However, in the first period, the differences in $E_{LUC}$ estimates may also partly arise from the discontinuity in the cropland area (Figure A2). In the 1940s, even though $E_{LUC}$ simulated by ORCHIDEE-MICT remains within the (large) inter-model range of the TRENDY models, the two land-cover datasets FSU-AGR and LUH result in substantial differences in the estimated $E_{LUC}$. These differences do not result from FSU–NEW indicating systematically lower crop area, since they are not present in the previous decade. Rather, the lower $E_{LUC}$ are mainly due to the sharp decrease in cropland area during 1940-1943 and the resulting increase in natural vegetation growth.

Considering average reference values of $CO_2$ uptake by different forest types (Luyssaert et al., 2007) and assuming that the decrease in cropland extent between 1940 and 1942 was replaced by young growing trees proportionally for each PFT, the estimated net $CO_2$ uptake resulting from forest expansion over abandoned cropland would be of about 0.09PgC.yr$^{-1}$. This value is consistent with the differences in net $CO_2$ uptake relative to the reference simulation in the FOR scenario of 0.07PgC.yr$^{-1}$. The difference for the grassland scenario is higher, ca. 0.12 PgC.yr$^{-1}$, since grasslands are more productive than forests in the model (Table A1) and grasses allocate more carbon to belowground biomass. As there was an increase in cropland area from 1944 onwards, the resulting differences in $E_{LUC}$ over the decade are attenuated. However, the simulations using FSU–NEW data still lead to values of emissions from LUC 45% to 78% lower than the one using FSU–REF during the 1940s. Over the full decade, the decrease in cropland land during the peak of WWII contributes to an increased sink of 0.04-0.07PgC.yr$^{-1}$, i.e. 6-10% of the additional sink required to explain the discrepancies between observations and recontructions of atmospheric $CO_2$ during the *plateau* (Bastos et al., 2016).

In the 1940s, very high temperature anomalies were registered in FSU, over a latitudinal band that is predominantly covered by grassland in the west and forest in the east (Figure A1), and where vegetation productivity is temperature limited (Greve et al., 2014). Therefore, it is expected that the high warming persisting over a decade might have been responsible for increased vegetation growth and higher $CO_2$ uptake. Indeed, increased tree-ring width in high-latitude locations in the Northern Hemisphere, and particularly in Eurasia, has been reported (D'Arrigo et al., 2006), adding confidence to our results.

Such processes are likely better represented in ORCHIDEE-MICT, which is specifically developed to represent the specific dynamics of soil carbon and vegetation productivity in high-latitudes, than in the other TRENDY models. Indeed, ORCHIDEE-MICT simulates a strong increase in vegetation growth in those regions experiencing warming, resulting in strong $CO_2$ uptake and also C accumulation in soils (even if the warming effect leads to lower soil C accumulation than in $S_{Clim}$).

The comparison of $S_{Ref}$ and $S_{Clim}$ shows a strong enhancement of C fixation in vegetation and generally a decreased capacity of C fixation in soils in response to the higher temperatures registered in the 1940s. While the TRENDY models do show a decrease in soil C during 1940s, none is able to capture such response. The net $CO_2$ uptake by natural vegetation simulated by ORCHIDEE-MICT in the 1940s is 0.4PgC.yr$^{-1}$ higher than any of the land-surface models, which could explain ca. 60% of the additional sink required in Bastos et al. (2016) (considering the gap of 0.7PgC.yr$^{-1}$ estimated by DGVMs).



## 6 Conclusions

Here, we attempt to evaluate the hypothesis proposed in Bastos et al. (2016), that the economic and social disruption imposed by WWII could have resulted in lower emissions from land-use change during the 1940s, partly explaining the stabilization in atmospheric $CO_2$ during this decade (*plateau*). Their rationale was that massive and abrupt cropland abandonment could

have been followed by carbon stock recovery in many the FSU regions affected by war, and therefore result in an increased terrestrial sink. Such short-term fluctuations are not currently represented in the datasets used to estimate emissions from LUC, as they are developed with the main purpose of estimating $E_{LUC}$ over centuries (Hurtt et al., 2011).

In order to evaluate the hypothesis, we compiled a new dataset of cropland area in the Former Soviet Union, based on annual economic reports between 1913 until 1961. It should be noted that while the agricultural output compiled in Soviet national

statistics has been questioned, the area occupied by agriculture is likely reliable (Nove, 1982; Davies et al., 1994). Our dataset shows good agreement with other social and economical statistics, indicating that the changes in cropland area reported in our dataset are realistic.

We find that the massive reduction of cropland area between 1940 and 1943 (62 Mha) leads to differences in $E_{LUC}$ of 0.07-0.12PgC.yr$^{-1}$ relative to the reference simulation during these few years and affect considerably the decadal average NBP. Even

if the reduction in $E_{LUC}$ due to land-abandonement in this decade corresponds only to a small fraction of the sink gap during the plateau period (6-10%), it is likely that decreases in agricultural areas during WWII might have also occurred in other regions and could additionally contribute to the sink gap. Such events may not be included in FSU–REF data due to the use of different sources of information as is the case, for example, of China (He et al., 2013)). Furthermore, here we show that extreme but relatively short LUC events (such as the land-abandonment reported here) may have an impact on the decadal resulting $CO_2$

fluxes which are not accounted for by time-series that represent decadal variations, highlighting the relevance of the temporal resolution in LUC datasets.

The Northern Hemisphere warming during the 1940s discussed here and in Bastos et al. (2016) is a consistent feature in observation-based and proxy data (Mann et al., 1999; Compo et al., 2013). Our simulations indicate a very strong enhancement of net $CO_2$ uptake by 0.4PgC.yr$^{-1}$, vegetation growth and accumulation of carbon in soils in the mid- to high-latitudes coincid-

ing with warming, which is supported by tree-ring data (D'Arrigo et al., 2006). The fact that other land-surface models do not seem to capture such response highlights the importance of correctly representing and parametrizing high-latitude processes to capture the effects of warming on boreal vegetation. This climate induced enhancement of NBP in the FSU territory accounts for about 60% of the global sink gap of the plateau period.

Our results indicate that tree growth in response to the high-latitude warming registered during the *plateau* period may likely

contribute to a greater extent to the required $CO_2$ sink discussed in Bastos et al. (2016) than land-use changes. However, the contribution from LUC is still non-neglibile, especially if extended to other regions.

Further efforts are needed to better understand the extent of tree growth in response to the 1940s warming, which appears to be supported by the tree-ring record (D'Arrigo et al., 2006). At the same time, such information is valuable to identify key processes that need to be improved in land-surface models in order to better represent decadal variability in the land $CO_2$ sink.





*Data availability.*   The data on total and regional agricultural area and population in FSU are provided in Supplementary Data.

*Author contributions.*   A.B. conducted the analysis, performed the simulations, prepared the figures and wrote the manuscript. A.P. and S.K. collected and organised the new dataset. E.A.G. and C.G. performed the updated of the land-cover maps. C.Y. and W.L. helped performing the model simulations and conducting the analysis. P.C. provided expert advisory during the conception and development of the study. A.P.,
5   W.L., C.G., C.Y. and P.C. contributed to the writing of the manuscript.

*Acknowledgements.*   The work is supported by the Commissariat à l'énergie atomique et aux énergies alternatives (CEA). The authors would like to thank the TRENDY modelling teams and S. Sitch and P. Friedlingstein for providing and maintaining the TRENDY outputs. We thank the valuable help of Dan Zhu, Fabienne Maignan, Shushi Peng & Albert Jornet during the development of this study.



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





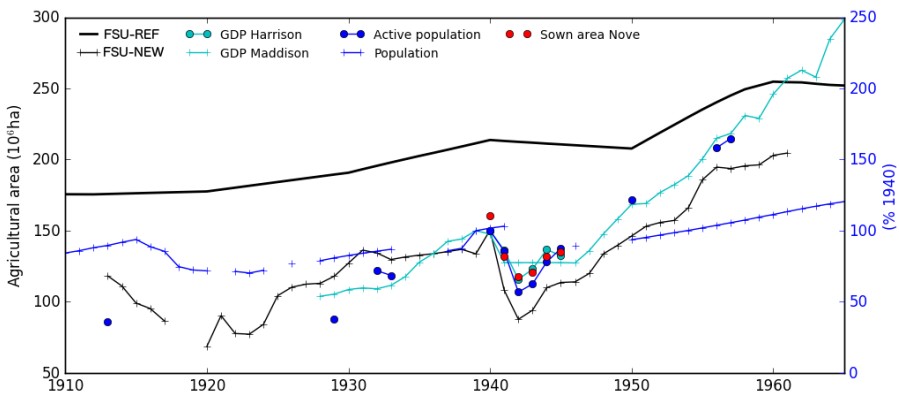

**Figure 1.** Agricultural area in the FSU in the two datasets used here: FSU–REF (plain line, black) and FSU–NEW (black line with markers). Gross domestic product from two sources of data (light blue lines, cross markers indicate data from Maddison (2001) and circles from Harrison (2000a)) and total and active population collected from national statistics (blue, from Supplementary Data). The left $yy$-axis refers to the two agricultural extent datasets, while the right $yy$-axis refers to the socioeconomic statistics, which are presented as % of the 1940 reference value.





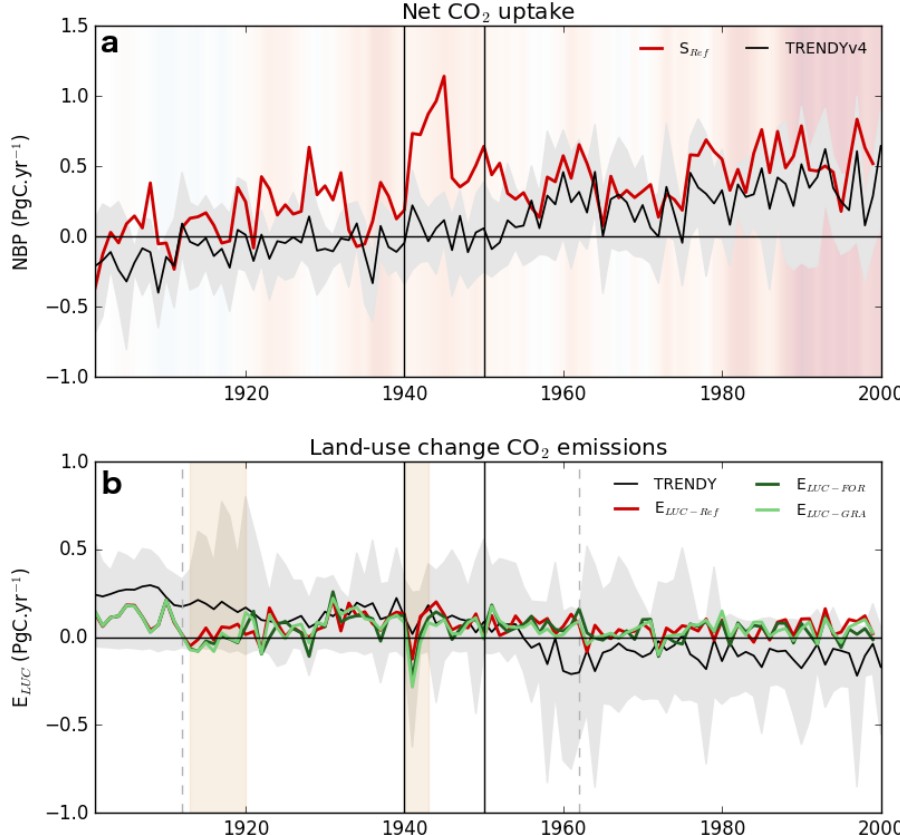

**Figure 2.** Simulated terrestrial $CO_2$ fluxes (NBP) in FSU between 1900-1999: **(a)** terrestrial natural sink (NBP) from $S_{Ref}$ and **(b)** emissions from LUC calculated by the difference in model simulations with LUC from FSU–REF and FSU–NEW and $S_{noLUC}$. By convention a land sink has positive NBP (flux from the atmosphere to the land), while positive emissions correspond to flux from land to the atmosphere. The fluxes simulated by ORCHIDEE-MICT are compared with the fluxes estimated by models in the TRENDYv4 intercomparison (Sitch et al., 2015) (bold black line represents the inter-model mean and the shade the inter-model spread), which are used as reference values in (Bastos et al., 2016). The vertical lines indicate the period corresponding to the new cropland area data (dashed) and the *plateau* period (1940-1950, solid black). In **(a)** the shaded colors indicate the anomaly of average annual temperature (from CRU/NCEP) over FSU relative to the 1901-1930 mean, with red colors corresponding to warming. In **(b)** the yellow shades indicate the periods of strong decrease in cropland area reported in FSU–NEW.





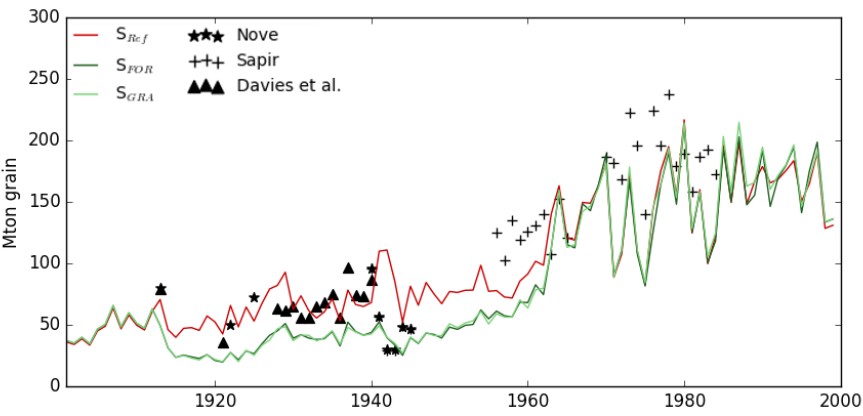

**Figure 3.** Comparison of simulated agricultural production in FSU over the 20<sup>th</sup> century using the two land-cover datasets with economical data. The harvest C flux was converted to million tons of grain for $S_{Ref}$ (forced with FSU–REF, in red) and $S_{FOR}$ and $S_{GRA}$ (different green shades), both forced with FSU–NEW vegetation cover maps.



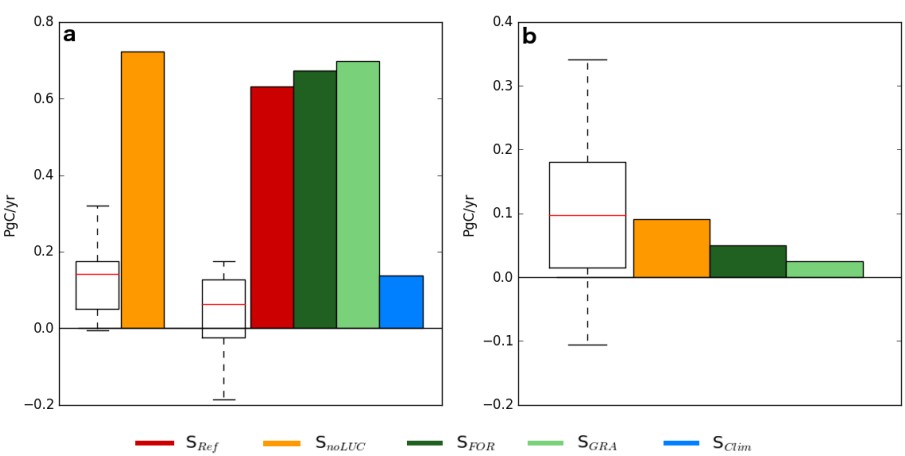

**Figure 4.** Simulated $CO_2$ fluxes during the 1940s over FSU: **(a)** net land-atmosphere flux (NBP) and **(b)** land-use change emissions. The color bars indicate the fluxes simulated by ORCHIDEE-MICT according to the legend, and the boxes show the corresponding distribution of NBP and $E_{LUC}$ simulated by the TRENDY models. TRENDY S2 is compared with $S_{noLUC}$ and TRENDY S3 with the other four simulations, all including LUC.





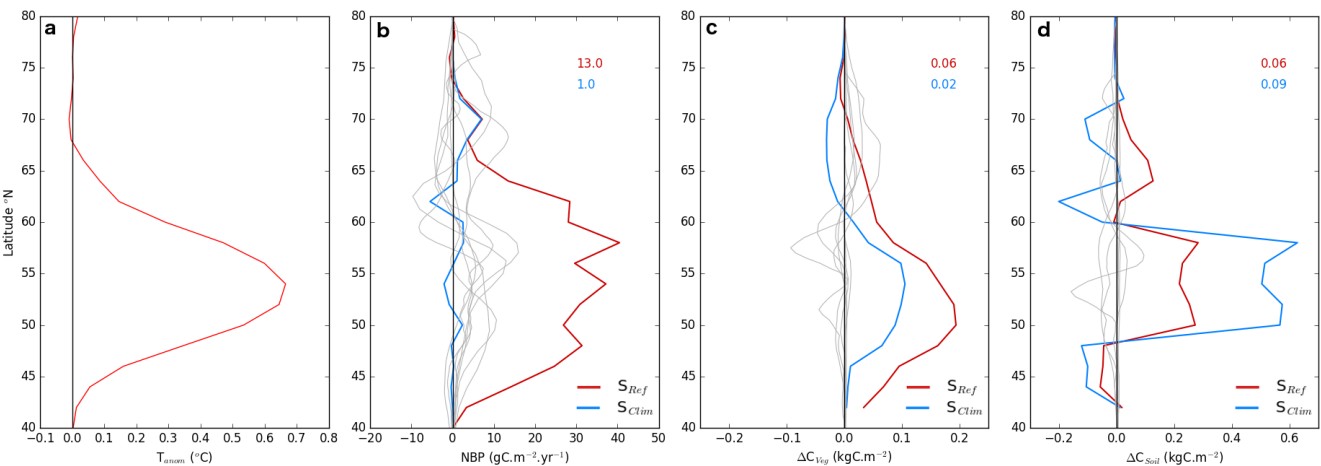

**Figure 5.** Latitudinal differences between the 1940-1949 and 1930-1939 over the FSU region, for **(a)** surface temperature from CRU/NCEP, **(b)** net CO$_2$ uptake (NBP), **(c)** carbon stocks in vegetation and **(d)** in soil. The two simulations with historical (S$_{Ref}$) and cyclic (S$_{Clim}$) climate variability are compared with the corresponding variables from the TRENDY models (shown individually in thin grey lines).





**Table 1.** Summary of the ORCHIDEE-MICT factorial simulations performed in this work and the corresponding hypethsis to be tested by each simulation. In $S_{Clim}$, the climate forcing is cycle over the same 10-yr period as the one used for the spinup (1901-1910).

| Simulation | Climate | LU map | Test |
|---|---|---|---|
| $S_{Ref}$ | Historical | FSU–REF | - |
| $S_{Clim}$ | Cyclic | FSU–REF | Climate |
| $S_{noLUC}$ | Historical | FSU–REF 1860 | $E_{LUC}$ |
| $S_{FOR}$ | Historical | FSU–NEW FOR | FSU–NEW crop → Forest |
| $S_{GRA}$ | Historical | FSU–NEW GRA | FSU–NEW crop → Grassland |





**Table 2.** Comparison of simulated data on C fluxes and stocks with literature values. Values are in PgC.yr$^{-1}$ for carbon fluxes (NPP, net sink), and in PgC for C-stocks (soil and biomass C). The dataset in Guimberteau et al. (2017) is calculated from two distinct observation-based datasets for "present day" period, therefore we compare it with the last year of our simulations.

| Variable | Region | Period | $S_{Ref}$ | $S_{FOR}$ | $S_{GRA}$ | Observation | Ref. |
|---|---|---|---|---|---|---|---|
| NPP | Russia | *early 1990s* | 4.0 | 4.2 | 4.2 | 4.4-4.5 | (Kudeyarov, 2000) |
| Net C sink | Russia | *early 1990s* | 0.5 | 0.6 | 0.6 | 0.8-1.0 | (Kurganova et al., 2010) |
| Soil C | FSU | 1999 | 846 | 836 | 853 | 709 | (Guimberteau et al., 2017) |
| Soil C | Russia & Ukraine agric. | 2000 | 6.8 | 8.4 | 9.7 | 8.8 | (Smith et al., 2007) |
| Vegetation C | Russia Forests | 1993 | 46.1 | 45.2 | 45.0 | 43.2 | (Shvidenko and Nilsson, 2003) |
| Vegetation C | FSU | 1993-1999 | 44.7 | 49.6 | 44.7 | 43.8 | (Liu et al., 2015) |

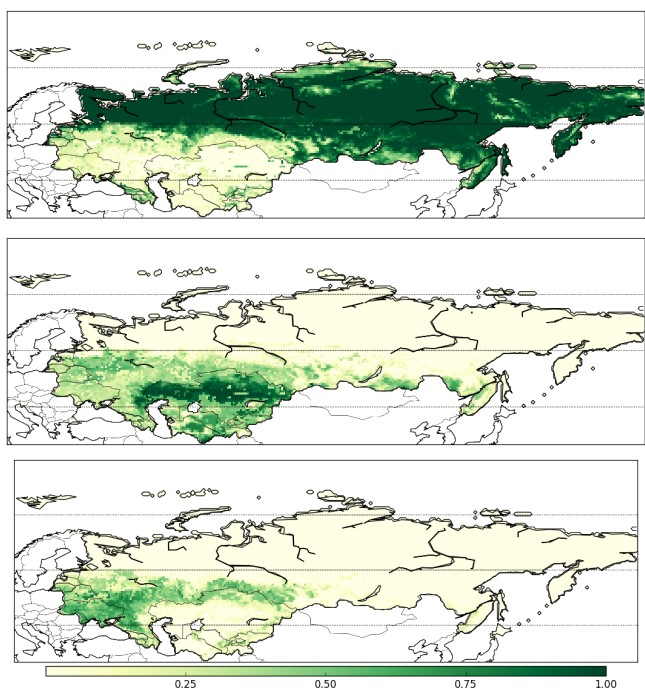

**Figure A1.** Average spatial distribution of forest (top panel), grassland (central panel) and cropland (bottom) area fractions in FSU during the 20$^{\text{th}}$ century (1901-2000 average), converted from FSU–REF to ORCHIDEE PFT classes.




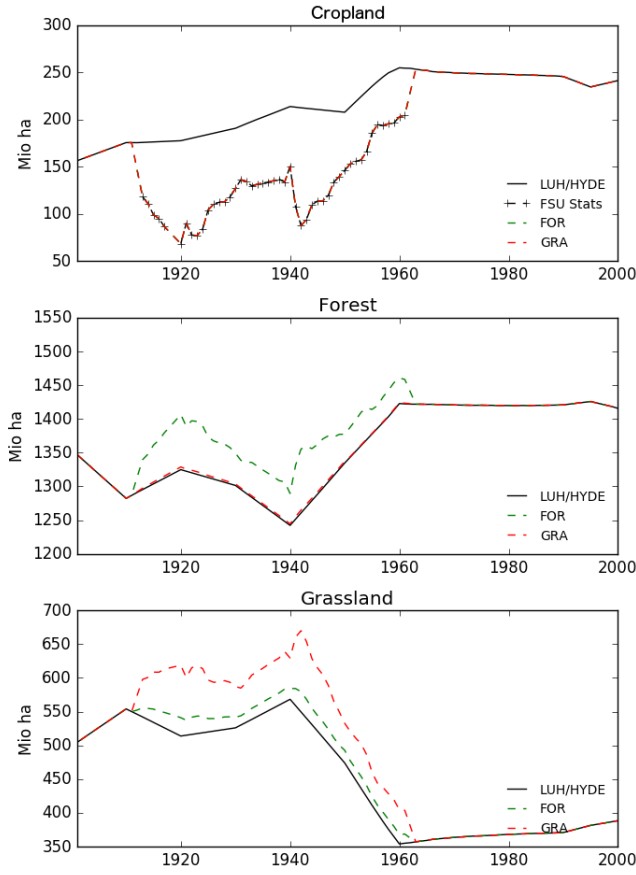

**Figure A2.** Evolution of land-cover classes in FSU during 1901-2000, for FSU–REF, FSU–NEW and updated values. Cropland area in the FSU in the two datasets used here: FSU–REF (plain line, black) and FSU–NEW (1913-1961, markers). The corresponding updated values in the corrected land-cover maps used in the model simulations are shown in dashed lines, differing from FSU–REF between 1912 and 1962 (FSU–NEW period and interpolation of 1912 and 1962 values to produce a smoother transition). In panels b) and c), the total FSU forest and grassland in FSU–REF and in the two updated datasets corresponding to each scenario: FOR (forest replacing cropland when possible, green) and GRA (grassland replacing cropland, red).





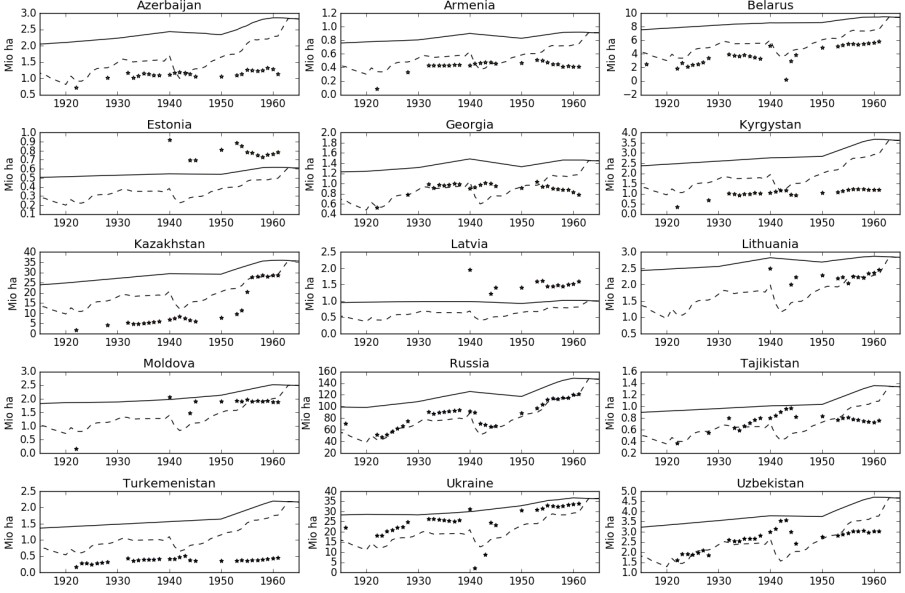

**Figure A3.** : LUC reconstruction for former republics of FSU now independent countries: comparison of FSU–REF (solid lines) and FSU–NEW (black markers) cropland area, and the corresponding original datasets used to reconstruct the FSU–NEW LUC (dashed lines).

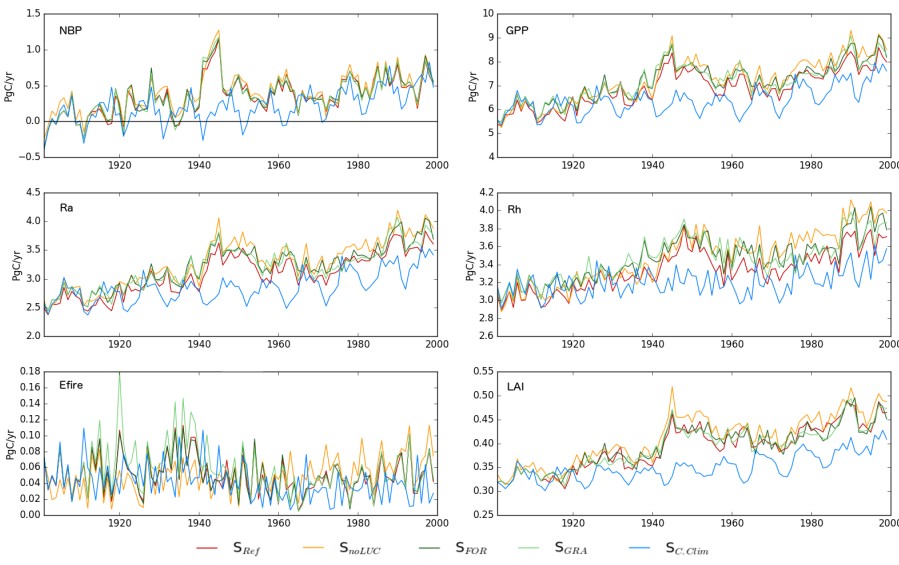

**Figure A4.** Main land-atmosphere fluxes and leaf-area index over FSU during the 20[th] century simulated by each of the five factorial experiments.



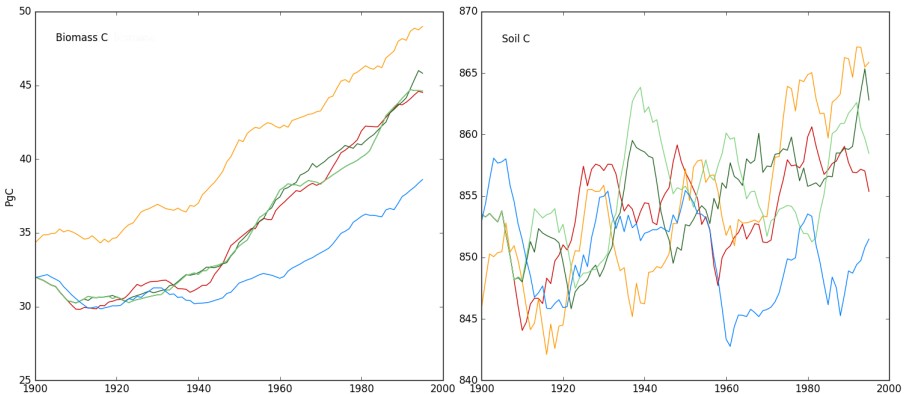

**Figure A5.** Simulated carbon stocks over FSU during the 20[th] century for each of the five factorial experiments.

**Table A1.** Plant functional types used in ORCHIDEE-MICT simulations, and corresponding Vc$_{max}$, modified from the standard setup for crop PFTs.

| PFT | Vc$_{max}$ (micromol.m$^{-2}$.s$^{-1}$) | 1901-1999 area USSR (Mha) |
|---|---|---|
| 1: bare soil | - | - |
| 2: tropical broad-leaved evergreen tress | 65 | 0 |
| 3: tropical broad-leaved raingreen trees | 65 | 0 |
| 4: temperate needleleaf evergreen trees | 45 | 116 |
| 5: temperate broad-leaved evergreen trees | 45 | 22 |
| 6: temperate broad-leaved summergreen | 55 | 133 |
| 7: boreal needleleaf evergreen trees | 45 | 326 |
| 8: boreal broad-leaved summergreen trees | 45 | 569 |
| 9: boreal needleleaf summergreen trees | 35 | 224 |
| 10: C3 natural grass | 55 | 429 |
| 11: C4 natural grass | 25 | 11 |
| 12: C3 agricultural grass | 50 | 185 |
| 13: C4 agricultural grass | 40 | 4 |