# Peer review of "The contribution of land-use change versus climate variability to the 1940s CO2 plateau: Former Soviet Union as a test case"

_Biogeosciences, 2017_

## Author Comment (AC1) · 24 Jul 2017

The authors thank the referee for accepting to review our manuscript. In response to the referee, we would like to stress some points mentioned in the manuscript that the referee might have overlooked (we acknowledge we can and will make them clearer in a future revision), and to make some notes regarding what is possible to evaluate with the model, and what is not. Please find below a point-by-point reply to each of the comments raised by the referee (attached a PDF with more legible text and the figures organized along the text):

RC1: Some confusion caused by the statement of the research problem. Right, CO2 concentration was not increasing during 1941-1945(46), however since 1947 steady constant growth already started (see NASA data https://data.giss.nasa.gov/modelforce/ghgases/Fig1A.ext.txt). But within the same period and up to the 1950-1960s the global average temperature didn't grow either. So, there is no any contradiction here. It is known that at cooler conditions CO2 dissolution in the ocean is increasing. AC1: We would like to call the referee attention to the fact that the research problem in this paper is not to consolidate that there is a missing sink. This has been thoroughly addressed in the companion paper Bastos et al. (2016), which is cited in the second sentence of the abstract (and profusely throughout the manuscript). The goal of this work is to evaluate the contribution of two of the processes hypothesized by Bastos et al. (2016) to explain the missing sink. Nonetheless, we would like to stress that the issue is more complex than a simple increased dissolution effect as the referee implies, as discussed deeply in the companion paper, and summarized below. In fact, to the best of our knowledge, no work has proposed a global-temperature effect on the atmospheric CO2 stabilization during this period. On the contrary, several works (as discussed in Bastos et al., 2016) have tried to evaluate the robustness of this stabilization and attribute it to the ocean and terrestrial sinks, but they reach very different conclusions depending on the methods and assumptions used to calculate atmospheric CO2 growth rate (AGR) from CO2 concentration measurements (Joos and Bruno, 1998; Joos et al., 1999; Trudinger et al., 2002). Secondly, air trapped in the bubbles within an ice-core core corresponds to a mix of air with different ages, and therefore CO2 concentration in a giver year will be a mix of somewhat older and somewhat younger air, with the age mix being defined by the rate of accumulation of snow in a given place. Since measurements from different ice-cores are scattered in time (and space) and in order to produce one single dataset as the one that the referee indicates (which is consistent with the one in Bastos et al. (2016)) a spline needs to be fit and therefore, assumptions need to be made about the age effects and the uncertainty of the measurements (for a thorough discussion see Trudinger et al., 2002). Actually, Bastos et al. (2016) and many previous works (e.g.

Enting et al., (2006)) have addressed this problem thoroughly, for example as shown in Fig. S1 in Bastos et al. (2016), shown in Fig. 1. Because of age-mixing effects, it is not fully correct to assume (as the referee proposes) that the end of the stabilization period is in 1947, since the difference between 1946 and 1947 is only 0.1ppm (310.3 to 310.4). In fact, Bastos et al. (2016) have addressed the issue of the robust detection of the plateau timing (and you may note that they also discuss the timing proposed by previous works) by performing a structural-change analysis of the CO2 concentration trends (to exclude non-significant variations), and identify 1940 and 1950 as significant trend-break points (figure 2 here, figure S2 in Bastos et al. (2016)). Finally, as Bastos et al. (2016) demonstrate using simulations from Earth System Models, the ocean response only is not likely to produce a sink that equals CO2 emissions in that period. The referee may note that this is mentioned right in the abstract: "Their study indicates that even considering an enhancement of the ocean sink, still a gap sink of 0.4-1.5PgC.yr-1 in terrestrial ecosystems is needed to explain the CO2 stabilization."

RC2: As for the average temperature – please, see attached Fig.1 Upper line – average temperature of Russia, lower – world temperature. As we could see, during 1940-1950s in Russia the decrease of the average temperature was more than for the world. AC2: The authors would be pleased to see the axis on the figures, otherwise it is not easy to identify the periods of warming and cooling. Nonetheless, the authors believe the referee may be mixing the concepts of temperature anomaly (warmer/cooler than reference conditions) as discussed in our manuscript and temperature trends. The period 1940-1950 was characterized by a warm anomaly relative to the previous decades (1901-1930) both globally and in the Northern Hemisphere, as can be seen in the HadCRUT4 dataset (see figure 3)

Indeed, in our manuscript, we also show that warm temperature anomalies at decadal time-scale were observed over the FSU territory, which justifies our study focused on this region of the Northern Hemisphere. Please note the color shading in the background of Figure 1a (reproduced here as Figure 4), showing temperature anomalies

relative to the period 1901-1930 (red/orange warm anomalies, blue cold anomalies):

Indeed, the peak of the temperature anomaly seems to occur around 1938, with temperature decreasing over the following years, but nevertheless, the decadal temperature anomaly is indeed positive. This is relevant because vegetation will respond to the temperature anomalies more than the sign of the trend: e.g. the onset of the growing season is more likely to depend on whether the spring temperature is above a given threshold, than on whether one year is slightly cooler than the previous (provided T is still above the threshold). We would, therefore, like to stress that in our manuscript we analyze temperature anomalies, and not temperature trends.

RC3: Looks strange that as sources for national soviet statistics authors used foreign publications of (Nove, 1982; Sapir, 1989; Davies et al, 1994) instead of original statistical data. Such data should be available in the State archive of the Russian Federation. It might be recommended to the authors to check original data of the Central statistical office of the USSR and Central administration of national economic statistics of Gosplan of USSR. Without that it is difficult to judge on the accuracy of the activity data. AC3: As mentioned in Section 2.1.1 (Russian and Soviet crop area), our new dataset of FSU agriculture area is collected from official national FSU statistics from Lyuri et al. (2010). National statistics were provided for the Former Soviet Union (FSU) during the period 1917-1961, and the Russian Empire during the pre-Soviet period starting from 1913 (dataset and full list of references may be found in Supplementary Data). The referee may revisit the Supplementary Material, where we present the full list of references (22 total) from where the data was collected, all from Soviet and Russian Empire statistics, including the Gosplan of USSR. The referee may also note that the international works cited are important references in the study of Soviet economy and that their studies were also based on soviet statistics (as the referee may find in their manuscripts). The reason to include such studies is precisely to show that, even if their values are somewhat different, they agree better with our newly collected dataset than with the LUH/HYDE (which is also based on international data). Given the high

uncertainty in evaluating changes in land-use in the past (which is discussed in lines 18-29 in Page 3 of our manuscript), our perspective is that having alternative sources of data, rather than being a weakness, adds robustness to our study.

RC4: One of the crucial omissions of the authors could be their assumption on fire emissions. On the page 6 it is stated that "fire occurrence is simulated using the SPITFIRE fire model ... which is well calibrated to simulate boreal fires". ....???? Authors have not estimated any $CO_2$ emissions from other burning processes during the WW2. It is well known that annual burned area of abandoned fields, forests, even villages and whole cities on the occupied territory should be large. Fire occurrence and burning of biomass within the front line should be even more essential. Disregard of the other sources for biomass burning leads to potential significant underestimation of $CO_2$ emissions in the research. AC4: Regarding the ability of the model to simulate boreal fires, the referee may revisit the reference provided in the same sentence (Yue et al., 2014) and the model evaluation paper (Guimberteau et al., 2017). Here with the prognostic fire model our focus is to include the open biomass burning in the carbon balance accounting. Yue et al (2014) Figure 10 showed the simulated burned area agrees well with another data source based on national statistics on a decadal basis over the 20th century (see Figure 5). Guimberteau et al (2017) showed that our simulated fire carbon emissions over the boreal Asia agree well with the GFED4s data set based on satellite-derived burned area over 1997-2009, confirming well the model capacity to simulate open biomass burning. We agree with the referee that fire emissions in our work are likely underestimated because of human-made fires during war conditions not being modeled. However, given the difficulty of making studies so back to the past with so little information, we try to keep our work the least speculative possible and therefore fires are included in a "baseline" scenario. However, if the referee could point us to referenced datasets on annual burned area in the war front, we would gladly attempt to incorporate this in our analysis. Nevertheless, we acknowledge that further discussion should be added in the revised version of the manuscript about the possible underestimation of fire emissions in the current setup of the simulations.

RC5: There is no any assumption in the research on the disturbance of the natural vegetation during the war, as well as for abandoned fields. For sure, we should assume not only its regular burning, but also losing of the soil layer totally with craters from the bombs and trenches. As a difference between the war on the FSU territory and Europe – most territory left occupied after fighting on the front line in FSU was pitted with holes. These are additional omitted $CO_2$ emissions. AC5: The authors agree with the referee that is it very likely that war resulted in relevant damages to forests, grasslands and agricultural fields. However, as in the previous point, we do not have any reliable data that could allow us simulating such effects. Therefore, our analysis is focused on: (1) the difference in $CO_2$ emissions due to land-use change estimated resulting from the use of the LUH/HYDE dataset (a reference in LUC studies, e.g., Le Quéré et al. (2015), Hansis et al. (2015), Stocker et al. (2014)) and our newly collected dataset and; (2) the influence of the climate anomaly on ecosystem $CO_2$ uptake and soil carbon stocks. Please note that even if such information would be available, the correct way to assess the impacts of war on ecosystems would be to perform two factorial simulations (one with war effects, one without). Because we focus on differences between pairs of simulations, both excluding such potential effects, they do not influence the results of our factorial simulations. We will introduce a discussion on this issue in the revised version of the manuscript.

RC6: It is questionable the correctness of the modeling of revegetation process by the model. If there is any verification of the model ORCHIDEE-MICT for abandoned croplands on the territory of Russia – such literature sources should be included and the accuracy of such modeling should be discussed. Overwise the model could not be applied for such kind of research. AC6: The authors acknowledge that the manuscript should include a reference to previous estimates of the impact of land-abandonment in the territory of Russia, even if performed for different times in history. Indeed, Kurganova et al. (2014) has performed an intercomparison of estimates of changes in carbon stocks after land-abandonment following the collapse of collective farming in Russia. In their study (table in Figure 6) they include the results from Vuichard et

al. (2008), which used a previous version of the ORCHIDEE model. The referee may note that this version of ORCHIDEE underestimated the rates of C sequestration, likely because the version of the model used lacked important soil processes which are included now in ORCHIDEE-MICT (a thorough description of the improvements to the model may be found in Guimbertau et al. (2017). Furthermore, as discussed in lines 7-12 in Page 12, our results for CO2 rates from vegetation regrowth in areas registering cropland decrease are consistent with values from literature. The authors will introduce some sentences regarding this point and the complete references in a revised version of the manuscript.

RC7: There are no data on the actual land use changes, only net cropland area change is available. However, the actual distribution of abandoned fields is crucial. Recent investigations of the process of revegetation on abandoned croplands in Russia (Romanovskaya, 2008 (in Russian)) show that croplands on the south (Rostovskaya obl., Krasnodarsky krai, Stavropolsky krai, the corresponding area of Ukraine) are losing soil C during first 5-7 years of the abandonment. During the WW2 all these territories were occupied and abandoned. Thus we should estimate emissions there, no sink. AC7: Indeed, our analysis starts from total cropland area change, however, as discussed in the Methods section 3.1, we produce spatially explicit annual maps, which are then used to force the model for changes in C pools and fluxes. As we explain in lines 24-28 in Page 6, in order to convert national total changes to pixel-level changes, we subtract the difference between FSU-REF and FSU-NEW proportionally to the fraction of cropland in each pixel. As an example, the map of the difference between the two datasets for the year 1940 is shown in the Figure 7 (from Reis, 2017), with number corresponding to fraction of pixel area:

"Indeed, most cropland change is located in the south-west. Nevertheless, the authors would like to point out that this issue is discussed in Section 5 (line 11-16, Page 11):

The method used here to update LUC maps does not account explicitly regional dynamics, such as the displacement of farmland from the front and occupied regions

during WWII to the eastern countries (Linz, 1984). We deliberately did not account for this displacement, as it would imply changing also natural vegetation fractions in other regions of the FSU (e.g. require forest/grassland removal), and increase the possible inconsistencies between the datasets. Even though such differences may be relevant at local scale, they are unlikely to significantly change our results for the aggregated FSU. Our data still provides a better match to country-level estimates of crop area (Figure A3) than previous reconstructions."

As for the resulting sinks from abandoned croplands, Kurganova et al. (2014) have made an review of different estimates of post-abandonment $CO_2$ exchanges in the Russian territory, and all studies indicate a total strong sink in the first 10-20 years after land-abandonment. This does not mean that in some regions, cropland abandonment might have led to a source. Indeed, our simulations indicate a small source resulting from land-abandonment in some southern regions (as shown in Figure 8), however this is largely offset by increased uptake in other regions.

Figure: Difference in NBP in 1940-1950 between the two factorial simulations SFOR and SRef. Values correspond to g.m2.decade-1.

RC8: It is not clear how revegetation into the forest was assumed in the modeling. During 5-7-10 even 15 years could not appear any forests on the abandoned fields. If we add to the assumptions regular burning of abandoned fields we could not estimate any C sink for young growing trees on that area. Thus, it could be potential overestimation of the C sink for that simulation as well. AC8: The assumptions and methods used to define each of the two scenarios (forest and grassland regrowth) are explained in Section 3.1 and 3.2, line 29 Page 6 to line 7 Page 7:

"Given that we have no additional information about forest or grassland changes in FSU during this period (apart from FSU–REF), we define two scenarios for the natural vegetation replacing crop area after abandonment. In the first one, crop area in each pixel is replaced by forest cover if forest is already present otherwise it is replaced by

grassland (FOR). The second scenario (GRA) is similar, but with grasslands replacing abandoned cropland, if grassland is already present, and otherwise allocated to forests. It should be noted that these two cases correspond to the two extremes of the possible range of forest vs. grassland trajectories in regions where agricultural area was abandoned. The resulting total forest and grassland areas over FSU corresponding to each case are shown in Figures A2b, c.

3.2 Model simulations The information about forest and crop and grassland fractional cover and transitions from LUH1 was converted to 2x2 degree lat/lon maps of the 13 PFTs in ORCHIDEE-MICT consisting of bare soil, 8 forest PFTs, 2 crop PFTs (C3 and C4 crops) and 2 grass PFTs (C3 and C4 grass). We consider only the region corresponding to the FSU, as highlighted in the shaded areas in Figure A1. The average PFT distribution over the 20th century in the FSU region, is shown in Table A1"

After a decrease in crop fraction in a given pixel, the resulting change is attributed to a forest PFT already present in the pixel (or a grassland PFT in the grassland scenario). This does not mean that a fully grown forest immediately follows land-abandonment, but that trees slowly start to regrow in this area, following a normal growth curve as in most land-surface models. As in our responses to RC4 and RC5, we try to avoid as much as possible assumptions not able to be validated by referenced data, therefore we do not think it is advisable to introduce the assumption proposed by the referee. As for the fire occurrence over the young forests generated from land abandonment, the referee added the regular burning assumption and argued that recovery carbon sink might not be that much considering fire emissions. However as far as we know there is little evidence showing that young reclaimed forests are more likely to burn than naturally regenerating forests with similar ages. On the contrary, young forests from former agricultural land would have smaller ground fuel than otherwise a naturally regenerating forest, and therefore are less likely to burn. One exception indeed exists however, that is active fuel wood collection over these forests on abandoned land. But as we explained in RC4, we focused on open biomass burning which our model is

capable of simulating. In general, both dendrochronological (Conard & Ivanova, 1997) and satellite-based studies (Giglio et al., 2013) indicate that Russian boreal forests are dominated by a fire return interval of 20–50 years up to 100 years, which is well captured by our model (Guimberteau et al., 2017, Figure 19). Based on this, we argue that the risk of underestimating fire emissions in recovery forests is low. We acknowledge, however, that one sentence about this issue may be added in the discussion in a revision of the manuscript.

REFERENCES

Bastos, Ana, Philippe Ciais, Jonathan Barichivich, Laurent Bopp, Victor Brovkin, Thomas Gasser, Shushi Peng, Julia Pongratz, Nicolas Viovy, and Cathy M. Trudinger. "Re-evaluating the 1940s CO2 plateau." Biogeosciences 13 (2016): 4877-4897.

Conard, Susan G., and Galina A. Ivanova. "Wildfire in Russian boreal forests—Potential impacts of fire regime characteristics on emissions and global carbon balance estimates." Environmental Pollution 98.3 (1997): 305-313.

Enting, I. G., C. M. Trudinger, and D. M. Etheridge. "Propagating data uncertainty through smoothing spline fits." Tellus B 58, no. 4 (2006): 305-309.

Giglio, Louis, James T. Randerson, and Guido R. Werf. "Analysis of daily, monthly, and annual burned area using the fourth‐generation global fire emissions database (GFED4)." Journal of Geophysical Research: Biogeosciences 118.1 (2013): 317-328.

Hansis, Eberhard, Steven J. Davis, and Julia Pongratz. "Relevance of methodological choices for accounting of land use change carbon fluxes." Global Biogeochemical Cycles 29, no. 8 (2015): 1230-1246.

Joos, Fortunat, Robert Meyer, Michele Bruno, and Markus Leuenberger. "The variability in the carbon sinks as reconstructed for the last 1000 years." Geophysical Research Letters 26, no. 10 (1999): 1437-1440.

Joos, Fortunat, and Michele Bruno. "Long‐term variability of the terrestrial and

oceanic carbon sinks and the budgets of the carbon isotopes 13C and 14C." Global Biogeochemical Cycles 12, no. 2 (1998): 277-295.

Kurganova, Irina, Valentin Lopes de Gerenyu, Johan Six, and Yakov Kuzyakov. "Carbon cost of collective farming collapse in Russia." Global change biology 20, no. 3 (2014): 938-947.

Le Quéré, Corinne, Roisin Moriarty, Robbie M. Andrew, Josep G. Canadell, Stephen Sitch, Jan Ivar Korsbakken, Pierre Friedlingstein et al. "Global carbon budget 2015." Earth System Science Data 7, no. 2 (2015): 349-396.

Reis, Érico Aboo Gani dos. Impactos das alterações do uso do solo nos fluxos de CO2 na União Soviética entre 1940 e 1960. MSc Diss. U.Lisboa, Portugal (2017).

Stocker, Benjamin D., Fabian Feissli, Kuno M. Strassmann, Renato Spahni, and Fortunat Joos. "Past and future carbon fluxes from land use change, shifting cultivation and wood harvest." Tellus B: Chemical and Physical Meteorology 66, no. 1 (2014): 23188.

Trudinger, C. M., I. G. Enting, P. J. Rayner, and R. J. Francey. "Kalman filter analysis of ice core data 2. Double deconvolution of CO2 and $\delta$13C measurements." Journal of Geophysical Research: Atmospheres 107, no. D20 (2002).

Vuichard, Nicolas, Philippe Ciais, Luca Belelli, Pascale Smith, and Riccardo Valentini. "Carbon sequestration due to the abandonment of agriculture in the former USSR since 1990." Global Biogeochemical Cycles 22, no. 4 (2008).

Please also note the supplement to this comment:
https://www.biogeosciences-discuss.net/bg-2017-267/bg-2017-267-AC1-supplement.pdf
* * *
[Figure]

**Fig. 1.**

[Figure]

**Figure S 2.** Piecewise linear regression model fit (orange dashed lines) on the annual values of atmospheric $CO_2$ between 1930 and 1960 (blue solid line) calculated from the spline-fit shown in Fig. 1. The trend break-points are marked by +, and correspond to the years 1940 and 1950. During this period, atmospheric $CO_2$ does not present any significant trend.

**Fig. 2.**

Northern Hemisphere

Southern Hemisphere

Global

HadCRUT4  Temperature anomaly (°C)

**Fig. 3.**

[Figure]

Fig. 4.

[Figure]

**Figure 10.** The annual burned area for 1901–2009 as simulated by ORCHIDEE (grey bar), reported by the Mouillot data (Mouillot and Field, 2005, black bar), and by GFED3.1 data (dashed white bar). Data are shown for the mean values over each decade for 1901–2000, and for 2001–2005 (2000 sA) and 2006–2009 (2000 sB). Refer to Sect. 2.4.1 for the correction of the Mouillot data by using GFED3.1 data.

**Fig. 5.**

**Table 3** Estimations of total carbon sequestration in former arable lands of Russia

| Period | Area (м ha) | Approach | Total C sequestration (Tg C) | Average rate of C sequestration (Mg C ha$^{-1}$ yr$^{-1}$) | Reference |
|---|---|---|---|---|---|
| 1990–2011 | 45.5* | Soil-GIS | 870 (254)* | 0.92 (0.28) | Present study |
| 1990–2011 | 45.5* | Approximation | 861 (646)* | 0.96 (0.72) | Present study |
| 1990–2006 | 30.2 | Soil GIS | 648 (47) | 1.26 (0.09) | Kurganova et al., 2010; |
| 1990–2006 | 30.2 | Approximation | 585 (33) | 1.14 (0.06) | Kurganova et al., 2010; |
| 1990–2005 | 27.9 | RothC model | 248 (37) | 0.55 (0.08) | Romanovskaya, 2008; |
| 1991–2000 | 20.0 | Orchidee model | 64 | 0.47 | Vuichard et al., 2008; ** |
| 1990–2004 | 34.0 | Approximation | 660 | 1.29 | Larionova et al., 2003 |

The estimations are based on different approaches and were done for various periods and areas of abandoned arable lands due to LUC after 1990. The one-sigma uncertainties of total C sequestration and average C sequestration rate estimated in the present study are shown in parentheses. The two studies at the bottom did not provide any uncertainty of the estimations.
*The rates of C sequestration and areas of abandoned lands were used for calculation of total C accumulation during the first 20 years after LUC (1990–2009). For the period 2010–2011 the total area of abandoned lands was 45.1 м ha and the average rate of C-sequestration was 0.19 Mg C ha$^{-1}$ yr$^{-1}$.
**This estimation included the whole area of former USSR based on FAO statistics for 2000.

**Fig. 6.**

[Figure]

**Fig. 7.**

[Figure]

NBP difference 1940-1950 Sfor-Sref

**Fig. 8.**

---

## Referee Comment (RC2) · A. A. Romanovskaya (Referee) · 25 Jul 2017

Dear Autors,

Thank you very much for very detailed explanations.

Please find my few additional comments below.

RC1: thank you. I think you are right.

RC2: I do not think there is a problem here, thanks. Only the data I would like to show - please, see the figure in the attachment. That is time series of spatially averaged anomalies of mean annual temperature at the earth's surface for Russia (top) and the

globe (bottom). Axis: horizontal - years, vertical - deviations from the mean during 1961-1990 (Celsius). Red shows the course of the 11-year average. That is a bit different from the curve for the Northern Hemisphere on your figure. So as for me I do not see for 1940s extra positive anomalies in Russia. Maybe it is even opposite - there is a strong negative anomaly in 1941.

RC3: thank you for the detailed explanation. That issue resolved.

RC4, RC5: In my personal view it is crucial. It is clear underestimation in your results. Unfortunately, I am not historican, and cannot help to find robust datasets for disturbances of ecosystems during the war. There are few data - for example, there are data on the number of burned villages (for example for Belarus it is about 9200 villages). Probable it is possible to find more indirect data in the Arhive of the Russian Federation, Ukraine, Belarus. Another way could be to obtain "expert opinion" - to find historican for that period. I do not think that just adding the discussion on the underestimation in the paper would be enough. We do not know the scale of that underestimation. In my view that could be very high and the results could be potentially misleading.

RC6: that the second point which is crucial. You have mention that ORCHIDEE model was verified and gave uncorrect results. And you assume that ORCHIDEE-MICT is now estimate correctly (?) I believe that the standart way of performing any modeling - is a verification against experimental data and assessing of the uncertainty of modelling in the beginning.

RC7: I tend to agree with you. Anyway some level of assumption and approximation should be applied. Thank you.

RC8: connected to RC4. Thank you.

Thank you very much again! Maybe it would be good to see opinions of another referees and editors.

[Figure]

[Figure]

Временные ряды
пространственно
осредненных аномалий
средней годовой
температуры у
поверхности Земли
для территории
России* и Земного шара**
за 1886-2014 гг.
Красным показан ход
11-летних средних

В среднем по территории
России самым теплым был
2007 год, за ним следуют
1995 и 2008 гг.

Для Земного шара в целом
самыми теплыми были:
2014, 2010, 2005 и 1998 гг.

* Данные "ФГБУ Институт
глобального климата и экологии
Росгидромета и РАН"

** Данные метеослужбы
Великобритании HadCRUT4.3.0.0
(http://www.cru.uea.ac.uk)

**Fig. 1.**

---

## Referee Comment (RC3) · A. A. Romanovskaya (Referee) · 25 Jul 2017

Yes, thank you.

I do not have problems with climate, I believe you have done that's correctly. That's a just may be a encouragement to clarify in the text about what you mean with the warming during 1940s.

Regarding war disturbances on lands: yes, I understood your point of comparing with the set of models in Bastos et al. (2016). However, the problem still exists in my view. You are introducing new estimates of C fluxes due to land use change, however, do

not take into account all disturbances (C losses) on abandoned land during the war. In that case problem only relates to the abandoned land. You underestimate C losses on abandoned land and therefore potentially overestimate sinks.

Verification - yes, of course, I read carefully your paper, Table 2 and comparison analysis. Still in my view, there is not comparison for abandoned lands, only total for Russia and FSU (stocks and C sinks). Your results - 0.4 PgCyr-1 - are still in the range of uncertainty between simulated data and estimations of net C sink with data of (Kurganova et al., 2010) in table 2.

---

## Short Comment (SC1) · 25 Jul 2017

Please find below a reply to the issues raised:

*RC2: I do not think there is a problem here, thanks. Only the data I would like to show - please, see the figure in the attachment. That is time series of spatially averaged anomalies of mean annual temperature at the earth's surface for Russia (top) and the globe (bottom). Axis: horizontal - years, vertical - deviations from the mean during 1961-1990 (Celsius). Red shows the course of the 11-year average. That is a bit different from the curve for the Northern Hemisphere on your figure. So as for me I do not see for 1940s extra positive anomalies in Russia. Maybe it is even opposite - there*

*is a strong negative anomaly in 1941*

AC2: The figure presented by the referee does not contradict our analyses (nor could it, since it is also based on the HadCRU temperature data, used to produce CRU/NCEP). Indeed there is a negative anomaly in one year (1941), but overall the 1940s decade is warmer than the period 1901-1930. Figure 1 shows the graph of annual and 5-yr mean temperature over the territory of the FSU (not just Russia) from CRU/NCEP (the dataset used to force our model). Again, as the referee points, 1941 is indeed colder than average (see Brönnimann et al., (2004) for an explanation of such cold anomaly), **but the decadal temperature anomaly is higher than any of the previous decades**. Also, the referee may verify the map of the temperature anomaly in the 1940s (Figure 2). Parts of western Russia might have been colder than average, but very large regions in high latitudes registered very warm anomalies (consistent with figures 2 and 5 in our manuscript).

*RC4, RC5, R8: In my personal view it is crucial. It is clear underestimation in your results. Unfortunately, I am not historican, and cannot help to find robust datasets for disturbances of ecosystems during the war. There are few data - for example, there are data on the number of burned villages (for example for Belarus it is about 9200 villages). Probable it is possible to find more indirect data in the Arhive of the Russian Federation, Ukraine, Belarus. Another way could be to obtain "expert opinion" - to find historican for that period. I do not think that just adding the discussion on the underestimation in the paper would be enough. We do not know the scale of that underestimation. In my view that could be very high and the results could be potentially misleading.*

AC: As mentioned in our previous reply, here we include the open biomass burning in the carbon balance accounting as a baseline estimate of fire emissions. Since our work is focused on the effects of (i) land-use change and (ii) climate variability, we perform factorial simulations to test the relative contribution

of these two processes in the resulting carbon budget. Therefore, unless the fires mentioned by the referee should significantly vary with any of the two processes (unlikely since they are human-caused), they do not affect our results (the underestimation is cancelled out). Furthermore, the referee may note that here we compare our results with the set of models in Bastos et al. (2016), and **none of these models includes war-related fires**. Therefore, even if we do acknowledge that the may some underestimation of the effects of fires, this does not significantly affect (i) the results from our factorial simulations and (ii) the comparison with TRENDY models. As mentioned before, **we avoid introducing processes in our analysis that are not based on reliable evidence**.

*RC6: that the second point which is crucial. You have mention that ORCHIDEE model was verified and gave uncorrect results. And you assume that ORCHIDEE-MICT isnow estimate correctly (?) I believe that the standart way of performing any modeling - is a verification against experimental data and assessing of the uncertainty of modelling in the beginning*

AC: As mentioned in our previous reply, a thorough description of the improvements to the model may be found in Guimbertau et al. (2017). The referee may note that **in their paper, the authors perform an exhaustive analysis of the ability of the model to simulate carbon exchanges, hydrological processes and soil carbon stocks, by comparing with several observation-based data**. In a revised version of the manuscript, a short summary of the model validation may be included. Nevertheless, the referee may note that in our Table 2, we compare the model simulated $CO_2$ fluxes and C-stocks with observation-based datasets Also, the referee is invited to revisit lines 32 (pg 8) to 9 (page 9) of our manuscript, where the

Bronnimann, S., J. Luterbacher, J. Staehelin, and T. M. Svendby. "Extreme climate of the global troposphere and stratosphere in 1940-42 related to El Niño." Nature 431, no. 7011 (2004): 971.

[Figure]

**Fig. 1.**

[Figure]

**Fig. 2.**

---

## Referee Comment (RC4) · Anonymous Referee #2 · 5 Sep 2017

The authors in the current paper (The contribution of land-use change versus climate variability to the 1940s CO2 plateau: Former Soviet Union as a test case) elaborate on their prior finding published in Biogeosciences (Re-evaluating the 1940s CO2 plateau doi:10.5194/bg-13-4877-2016) about the observation of the plateau of SOC emissions around 1940s and its possible drivers. One interpretation they find, or better to say, hypothesize, WWII venue and associated withdrawal of cropland from land use (62 Mha) across the former Soviet Union, could heavily contribute to such plateauing of the carbon emissions. On one hand, it is a very interesting hypothesis, but it has to be carefully validated. However, many statements and data used in this manuscript do not allow to validate such hypothesis. Presented study simplifies the process of

LUC caused by war conflicts, which in some cases may yield to massive abandonment (case of Nagorno-Karabakh http://link.springer.com/10.1007/s10113-014-0728-3 and Bosnia http://dx.doi.org/10.1080/01431160801891879 ), or opposite-maintaining extensive farming on occupied lands (feeding the Caliphate –ISIS in Syria and Iraq http://iopscience.iop.org/article/10.1088/1748-9326/aa673a ) (authors hypothesize all occupied land by Nazi was abandoned). Major issues here, A) authors do not account, not all Soviet agricultural land occupied by Nazis from 1941 to 1943 was abandoned (authors simply subtract numbers on cropland acreage from 1941 to 1943). Authors also did not consider the role of trade and food supply by the Soviet Allies (for instance, in the States we see cropland expansion over this period) and C emissions associated with deforestation during the war, fires and such and spatially differentiated land use displacement (it would be best at least to go with stats at the subnational level). B) Authors made a non-plausible assumption about immediate spontaneous afforestation (regarding the cropland extent Nazis occupied predominantly agricultural areas in forest-steppe and steppe zone, thus first would be grassland encroachment). The amortization period regarding C sequestration was not accounted in their model (ORCHIDEE-MICT modeling results were also not validated). C) Authors did not describe/ discuss why other socioeconomic shocks, such as the Civil War in Russia (30 Mha land was abandoned) and recent massive long-lasting abandonment after the breakup of the Soviet Union (60 Mha) did not reflect in the plateauing the C emissions. In the end of the abstract authors conclude "Even if land-abandonment during WWII might contribute to a relatively small fraction of the sink required to explain the plateau, it is still non-negligible, especially since such events have likely been registered in other regions". Given the all caveats with stats and assumption and expected abandonment would result in 6-10% of the gap sink required to explain the plateau (with current data), evidence suggest the role of abandonment was marginal in the plateauing C emissions and overstated.

I recommend to carefully evaluate all these caveats and to use a deductive approach to ground the hypothesis about the role of land-use change or simply concentrate on

the modeling exercise and leave out interpretation of drivers. I have a feeling, you just made a nice modeling exercise, and try to pin with the war story or better to say, the Soviet development until 1960 (it is also not clear, why until 1960?).You also never reflected on change in economy and industry during the war period and industry decline, which had probably higher contribution to reduction of C during the war in FSU compared to agricultural land use. It is also not clear from the text, if you also look just at the war period and contribution of WWII or the Soviet Union and associated land-use (you talk about war, you talk about soviet period, then about the period during the 20th century, however you tend to reconstruct stats from 190 to 1970?, figure 1), this does not help either to concentrate on your story. Last but least, you need a better structure of the manuscript; there are a lot of inconsistencies with the periods and data you used, the story is not clear, all sections of the manuscript are fuzzy and in the discussion you need to dedicate a fair amount of space to discuss your major findings, comparison with other studies and caveats.

Major issues/ caveats in more details. 1. Authors state from 1941 to 1943 abandonment comprised 64 Mha of croplands of the Soviet Union, which is one/ third of cultivated croplands in 1940, a pre-war year (they simply perform the subtraction of the numbers before and after the major peak of the Great Patriotic War in 1943 (name, which is common across the Soviet Union for the WWII activities at the Eastern Front) However, evidences suggest Nazi maintained agricultural production and reorganized kolhozes and sovkhoses to feed the Third Reich (12% of consumed food to Nazi Germany was coming from fSU), they had to feed Wehrmacht at the Eastern front (3 -4 mln people annually), plus to maintain remaining population on occupied lands (60% from total fSU's population, if i am not mistaken, were living on occupied lands). While the infrastructure has been deteriorated (50%-60% of agricultural equipment was not functioning by the end of 1941), deeply behind the Red's lines, particularly on Chernozem lands, the agriculture has been restored and even equipment was brought from Nazi Germany (this was also a part of Nazi's program to further expand at the expense of best endowed soils in Russia and Ukraine and resettle land poor Germans).

So, despite cleansing, massive deaths and destruction of equipment, Nazis still had to maintain agriculture to feed above-mentioned groups on occupied territories. I am not a big expert on the state of land use on occupied lands during WWII, but this has to be discussed and additional evidences are needed to estimate, which portion of occupied agricultural lands was truly abandoned. At the same time, a loss of agro-environmentally endowed lands forced Soviet Union partially expand production in Siberia and Kazakhstan- land use displacement (to capture such displacement you need to look at the oblast level statistics if such displacement occurred). Last but not least here, Soviet Union received technology and food support via lend-lease program from the States, thus causing via teleconnection additional cropland expansion in North America (please take a look at the agricultural statistics of the US). This has not been discussed and accounted in the C estimates, which makes me feeling, the contribution of Soviet Union in C savings is oversimplified. 2. Continuing here the discussion about the caveats, authors assume, a bulk of abandoned lands just in three years has reverted into shrubs and trees. First, depending on site condition, even in temperate Russia, it may take even up to 20 years for the field to be encroached by shrubs. At the same time, a bulk of potentially abandoned lands (occupied by Nazis) was located in the central and southern Russia and Ukraine (authors did not show an area of potentially occupied by Nazis). If Nazis would not maintain farming (which was not the case) in the Black soil region, first we would observe a grassland restoration, but not shrubs and trees. I also did not find how did you account where abandonment would occur (I see only stats at country level). 3. Studies confirm, during the first three years after the abandonment, there might be even C losses and largest C sequestration, particularly in the above ground biomass and in soil will start after 10 years after abandonment in European Russia and Ukraine (for further details please see Quandary over Soviet croplands http://www.nature.com/doifinder/10.1038/504342a and Schierhorn et al. 2013 http://dx.doi.org/10.1002/2013GB004654 ). In this regard, it is not clear how C gain just in three years has been assessed. Here I also provide a criticism; you also did not validate your modeling results on C uptake on abandoned lands, to reflect if

your estimates are plausible (no any soil chronosequences were used for the contrast and I did not see your C sequestration map for WWI period). 4. There have been similar massive LUC processes/ shocks-the Civil War in Russia (roughly 30 Mha were abandoned over similar period), post-Soviet transition from state-command to market driven economy (60 Mha from 1990 to 2010), Virgin Lands Campaign cropland expansion (just from 1953 to 1964-40 Mha on new croplands), recent massive afforestation in China. Why these processes did not yield to plateau or toward a rapid increase of C emissions? Reflection on that is necessary.

Additional comments. It would be best if you would provide line numbering to make the comments.

The abstract can be condensed.

It is not clear if data has been assembled at provincial (gubernya or oblast like level) or national level (one number per year for entire country). For instance, if we talk about the former Soviet Union, do you use only one value for Russia without further disaggregation for oblast (s) provinces? How then did you track, where did cropland expansion occur and where was an influence of WWII? Country-level data is not sufficient for such analysis, especially if you talk about land-use displacement (mentioned in the work of Linz).

Block 15. P 3 the Soviet Union is one of the well documented countries regarding the agricultural statistics (for instance, explicit statistics exists on crops and yields back to 1913). LUH1 report and Luyri et al. have different numbers, simply they use different assumptions about LUC. If statistics is scarce and contradictory to your opinion, why do you use the sources of contradictory statistics? Block 20 p 3 "the new farmland created likely did not compensate land abandonment in the affected war territories". I am questioning here, likely or did not? I would recommend by using a deductive method to further explore, how much, in fact, land was actually abandoned with account for land-use displacement. You use just one number per country, without any account

for land use displacement (you need to go down to oblast level data, which is available for FSU)

block 30 p. 3 not clear the objectives, if you talk about Soviet Union, or 20th century or WWII period if you reconstruct what happened during 20th century, then you need stats by 2000 (I see some simulations by 2000 and in figure 1 stats is between 1910 and1970,but in the text you state,1913-1916).

block 10 p.4 Lyuri et al. (2010) is a book in Russian, which chapter, or dataset did you use? How well Lyuri et al. data match Nove and which parameter did you use as a proxy for abandonment (arable land or sown areas?). I am a bit surprised you rely on assumptions from the book of Nove (1982) another edition is from 1990, which is a rough book for a broader audience. You need to disentangle land use and rely on studies on economic performance on occupied lands. There has been plenty of studies by Russian historians and former Generals of Wehrmacht (testimonies).

block 25 p. 4 how well LUH1 matches selected statistics from Lyuri and Nove ? it is not clear from the text why do you use Hurtt data jointly with HYDE 3.1. Plus earlier in the text you criticized HYDE data and then you use HYDE, I assume, to disaggregate your statistics at the national level.

2.2. Socio-economic statistics Why do you need population and GDP, how does this related to land-use demand and land abandonment. It would be best to explain in the introduction. How GDP for fSU is reliable?

Block 10 p.5 "The relationship between population and economic output with total crop area likely changed over the 20th century due to agricultural mechanization and fertilization or to rural exodus. Nevertheless, they provide reasonable proxies to evaluate the variability of crop area reconstructed by FSU–REF and the FSU–NEW statistics." Did you use linear relationship in the end between population and land-use demand? I fully agree, by the end of XIXth century, such linear relationship is not valid for Russia and fSU.

**2.3 ORCHIDEE-MICT**

Why did you use this model not any other process-based dynamic vegetation models (e.g., LPJ–GUESS? How did you (if you) validate your model, particularly, it was not designed to model C sequestration on abandonment. I assume, this model does not have agricultural component, such as LPJmL.

"The new soil carbon module was shown to reproduce the amount of 5 soil carbon in the high latitudes and the seasonal exchange of $CO_2$ resulting from the seasonal imbalance between gross primary productivity (GPP) and total ecosystem respiration (TER). Fire occurrence is simulated using the SPITFIRE fire model as described in Yue et al. (2014), which is well calibrated to simulate boreal fires". This all interesting, but how did you account for agriculture? It is not clear, how did you parametrize your model and what were inputs (e.g., crop rotation, mechanization, fertilizers, land use)? There is nothing sad if you validated your model, particularly on C stocks on abandoned lands.

**3.1 Updated gridded LUC data**

How did you account or simulate abandonment-prone area in the occupied zone? I do not see any reconstruction of land-use. Did you distribute evenly abandoned lands across fSU? 4.2 Carbon fluxes during the 20th century you tried to reconstruct land use from 1910 until 1960, right? How does this come to entire 20th century?

Block 10 p. 8 you use the references to almost yellow literature Linz and Nove, no any studies by the Russian historians or any other historians who worked on reconstruction of land use on occupied lands. Again, a subtraction of two numbers, before and after main venue of war, when a large portion of agricultural land was occupied, does not mean a complete termination of land use (since for Nazis it was a task and doctrine to obtain these fertile soils).

My elaboration on your modeling of C dynamic will be irrelevant based on the comments presented before, if you adjust them, they could change completely the picture

about sequestered C. I recommend carefully revisit my comments if you envision disentangle the effect of socio-economic shocks on land-use and C stocks. You may also downplay the story and concentrate on modeling exercise, with the account for above-mentioned comments.

————————————————————

---

## Author Comment (AC2) · 13 Sep 2017

We would like to thank the referee for reviewing the manuscript. We acknowledge that the clarity of the manuscript can be improved the relevance of the results better posed and are willing to revise the manuscript accordingly. Still, we believe the referee may have missed some of crucial aspects of this work related with: (i) the goal of this study; (ii)    the sources of the data used to update cropland area in fSU and their validity; (iii)     the spatial representation of the new LUC dataset;       (iv) the processes represented in the land-surface model and their credibility.

Regarding (i), this work is a comparative study focusing on two different processes that

may contribute to an increased biospheric sink as discussed in Bastos et al. (2016): natural climate variability versus land-use changes possibly unaccounted for in reference datasets (including the TRENDYv4 model results in Bastos et al. (2016)). Therefore, it is essential to compare our results to the DGVMs in TRENDYv4 (for climate variability) and with the LUC dataset used to force these models, i.e. LUH1, which is based on HYDE 3.1. These are **state-of-the-art models and datasets** used by the global carbon-cycle community to evaluate both processes (natural climate-driven C-sink and LUC emissions).

As for (ii), we pointed out that LUH1, since it is based on HYDE3.1, does not rely on national statistics before 1961 (since then, FAO data is available), but on a simple extrapolation based on country-level population to estimate changes in cropland area and uses a simple linear interpolation to produce annual values from decadal changes (Fig. 1). The use of total population in societies that underwent drastic socio-economic changes during the early 20th century (industrialization, rural-exodus) likely fails to reflect real changes in cropland area, especially during periods of drastic shocks (as the Civil War or WWII periods). Therefore, we made an effort to collect official statistics of cropland area from the Russian Empire and the fSU (reference list in Supplementary Data and at the end of this reply) until 1961, when global FAO records start (which is also based on national statistics). These official records are, to the best of our knowledge, the most reliable source of information. Several economists who studied intensively the fSU have discussed that while official crop production estimates have been questioned, the official numbers of cropland extent are considered reliable, as discussed in the manuscript (e.g. Wheatcroft and Davies, 1994). Naturally, as in any inventory, official numbers are subject to a certain degree of uncertainty, and this is even more true for early periods in history. But again, we would like to re-emphasize that we are interested in understanding how much could the differences between LUH1 and the national statistics we collected contribute to the estimated LUC emissions, as this is a comparative study. We believe such an explorative approach,
even though our collected data is also subject to uncertainty, can provide further insights on closing the carbon budgets for the period of 1940-1950.

(iii) Based on the new data collected for fSU totals (and country level), we produce spatially-explicit maps to force our model. These maps are updated based on the spatial distribution of cropland, forest and grassland in LUH1 and therefore take into account geographical differences in cropland distribution. These spatially-explicit maps essentially are based on satellite data, either the original LUC datasets (e.g. HYDE) or the ORCHIDEE reference PFT maps. We then **update** the gridded data in LUH1 on a pixel-by-pixel basis, distributing the differences between the two datasets at fSU level proportionally to the pixel-level fractional cover of cropland. This is common procedure in LUC studies (Peng et al., 2017). Even if the criterion is simple, the comparison of our updated maps with LUH1 at country level (Supplementary Figure A3) shows that our updated maps capture the country-level values reported in the national statistics, with especially good fit for the countries encompassing the largest fraction of total cropland extent. For a given pixel, a reduction of cropland is replaced by forest (FOR scenario) or grassland (GRA) plant-functional types. Again, we explain in the manuscript that this simple approach can provide two extreme scenarios that we further used to explore the carbon budget question, a typical approach in scientific investigation. In the land-surface model (iv), cropland is NOT immediately replaced by a fully grown forest nor does afforestation take place. It simply means that the model will simulate, after a decrease in cropland, forest-type (or grassland-type) vegetation slowly growing in place of crops, taking several years to reach maturity, and depending on climate conditions for growth and survival. The simulations we designed follow exactly the protocol used by the LUC community to estimate the legacy fluxes, loss of C-sink capacity and interaction with climate resulting from LUC, by using process-based models like ORCHIDEE-MICT (Houghton et al., 2012). This is common procedure for instance in the estimates of the Global Carbon Budget (LeQuéré et al. 2015), used in Bastos et al. (2016). Several factors cotribute to high uncertainty in LUC emission estimates (Pongratz et al., 2014), but

**our approach and model used are among the state-of-the-art methods used by the community** and are therefore, scientifically valid.

We address these issues in more detail in a point-by-point reply to the referee's comments in the PDF attached.

Please also note the supplement to this comment:
https://www.biogeosciences-discuss.net/bg-2017-267/bg-2017-267-AC2-supplement.pdf

**Supplement:**

**The contribution of land-use change versus climate variability to the 1940s CO2 plateau: Former Soviet Union as a test case Bastos et al. *Biogeosciences**

**Response to Referee #2**

We would like to thank the referee for reviewing the manuscript. We acknowledge that the clarity of the manuscript can be improved the relevance of the results better posed and are willing to revise the manuscript accordingly.

Still, we believe the referee may have missed some of crucial aspects of this work related with: (i) the goal of this study; (ii) the sources of the data used to update cropland area in fSU and their validity; (iii) the spatial representation of the new LUC dataset; (iv) the processes represented in the land-surface model and their credibility.

Regarding *(i)*, this work is a comparative study focusing on two different processes that may contribute to an increased biospheric sink as discussed in Bastos et al. (2016): natural climate variability versus land-use changes possibly unaccounted for in reference datasets (including the TRENDYv4 model results in Bastos et al. (2016)). Therefore, it is essential to compare our results to the DGVMs in TRENDYv4 (for climate variability) and with the LUC dataset used to force these models, i.e. LUH1, which is based on HYDE 3.1. These are **state-of-the-art models and datasets** used by the global carbon-cycle community to evaluate both processes (natural climate-driven C-sink and LUC emissions).

As for (ii), we pointed out that LUH1, since it is based on HYDE3.1, does not rely on national statistics before 1961 (since then, FAO data is available), but on a simple extrapolation based on country-level population to estimate changes in cropland area and uses a simple linear interpolation to produce annual values from decadal changes (Fig. 1). The use of total population in societies that underwent drastic socio-economic changes during the early  $20^{\text{th}}$ century (industrialization, rural-exodus) likely fails to reflect real changes in cropland area, especially during periods of drastic shocks (as the Civil War or WWII periods). Therefore, we made an effort to collect official statistics of cropland area from the Russian Empire and the fSU (reference list in Supplementary Data and at the end of this reply) until 1961, when global FAO records start (which is also based on national statistics). These official records are, to the best of our knowledge, the most reliable source of information. Several economists who studied intensively the fSU have discussed that while official crop production estimates have been questioned, the official numbers of cropland extent are considered reliable, as discussed in the manuscript (e.g. Wheatcroft and Davies, 1994). Naturally, as in any inventory, official numbers are subject to a certain degree of uncertainty, and this is even more true for early periods in history. But again, we would like to re-emphasize that we are interested in understanding how much could the differences between LUH1 and the national statistics we collected contribute to the estimated LUC emissions, as this is a comparative study. We believe such an explorative approach, even though our collected data is also subject to uncertainty, can provide further insights on closing the carbon budgets for the period of 1940-1950.

(*iii*) Based on the new data collected for fSU totals (and country level), we produce spatially-explicit maps to force our model. These maps are updated based on the spatial distribution of cropland, forest and grassland in LUH1 and therefore take into account geographical differences in cropland distribution. These spatially-explicit maps essentially are based on satellite data, either the original LUC datasets (e.g. HYDE) or the ORCHIDEE reference PFT maps. We then **update** the gridded data in LUH1 on a pixel-by-pixel basis, distributing the differences between the two datasets at fSU level proportionally to the pixel-level fractional cover of cropland. This is common procedure in LUC studies (Peng et al., 2017). Even if the criterion is simple, the comparison of our updated maps with LUH1 at country level (Supplementary Figure A3) shows that our updated maps capture the country-level values reported in the national statistics, with especially good fit for the countries encompassing the largest fraction of total cropland extent. For a given pixel, a reduction of cropland is replaced by forest (FOR scenario) or grassland (GRA) plant-functional types. Again, we explain in the manuscript that this simple approach can provide two extreme scenarios that we further used to explore the carbon budget question, a typical approach in scientific investigation.

In the land-surface model *(iv)*, **cropland is NOT immediately replaced by a fully grown forest** nor does afforestation take place. It simply means that the model will simulate, after a decrease in cropland, forest-type (or grassland-type) vegetation slowly growing in place of crops, **taking several years to reach maturity**, and depending on climate conditions for growth and survival. The simulations we designed follow exactly the protocol used by the LUC community to estimate the legacy fluxes, loss of C-sink capacity and interaction with climate resulting from LUC, by using process-based models like ORCHIDEE-MICT (Houghton et al., 2012). This is common procedure for instance in the estimates of the Global Carbon Budget (LeQuéré et al. 2015), used in Bastos et al. (2016). Several factors cotribute to high uncertainty in LUC emission estimates (Pongratz et al., 2014), but our approach and model **used are among the state-of-the-art methods used by the community** and are therefore, scientifically valid.

We address these issues in more detail in a point-by-point reply to the referee's comments.

RC1: The authors in the current paper (The contribution of land-use change versus climate variability to the 1940s CO2 plateau: Former Soviet Union as a test case) elaborate on their prior finding published in Biogeosciences (Reevaluating the 1940s CO2 plateau doi:10.5194/bg-13-4877-2016) about the observation of the plateau of SOC emissions around 1940s and its possible drivers. One interpretation they find, or better to say, hypothesize, WWII venue and associated withdrawal of cropland from land use (62 Mha) across the former Soviet Union, could heavily contribute to such plateauing of the carbon emissions.

**AR1:** Even though the referee's comments focus solely on the LUC aspect of this paper, but we would like to point out that the goal of this paper is to perform a comparative analysis of the potential contributions of two processes in terrestrial ecosystems proposed in Bastos et al. (2016) as mentioned in the abstract of the current manuscript:

They hypothesised that (i) the major socioeconomic and demographic disruptions during World War II (WWII) may have led to massive land-abandonment, resulting in an additional sink from regrowing natural vegetation which is not accounted for in most reconstructions and/or (ii) the warming registered at the same time, especially in the highlatitudes, might have led to increased vegetation growth and an enhancement of the natural sink.

And discussed in Sections 4.2 and 5. We acknowledge that the second aspect (climate effects on natural vegetation) should be elaborated in more detail in the introduction, so that the motivation of this study is clearer to the reader, which could also explain the choice of the land-surface model to perform the simulations (cf. RC5).

RC2: On one hand, it is a very interesting hypothesis, but it has to be carefully validated. However, many
statements and data used in this manuscript do not allow to validate such hypothesis. Presented study simplifies the
process of LUC caused by war conflicts, which in some cases may yield to massive abandonment (case of Nagorno-
Karabakhhttp://link.springer.com/10.1007/s10113-014-0728-3andhttp://dx.doi.org/10.1080/01431160801891879), or opposite-maintaining extensive farming on occupied lands

http://ax.aol.org/10.1080/014511608018918/9 ), or opposite-maintaining extensive farming on occupied lands (feeding the Caliphate –ISIS in Syria and Iraq http://iopscience.iop.org/article/10.1088/1748-9326/aa673a ) (authors hypothesize all occupied land by Nazi was abandoned). Major issues here, A) authors do not account, not all Soviet agricultural land occupied by Nazis from 1941 to 1943 was abandoned (authors simply subtract numbers on cropland acreage from 1941 to 1943).

**AR2:** We agree with the referee that evaluating changes in LU and their associated impacts, especially in a period where few data were available requires careful analysis. In the case of the fSU, the massive land abandonment registered during the war period is not a new hypothesis, and has been extensively documented in several economic and social studies, most of which are referred to in our paper. Because many of these works provide different numbers or only general trends (as the referee points out below), we sought carefully collected and harmonized national statistics data from fSU (and Russian Empire) for the period preceding 1961 (starting from which FAO data is available as mentioned in p2 line 26). The main point of this paper is that the data we collected match better the variations in agricultural area previously reported (e.g. during the Russian famine and Civil War as well as during the WWII period) than LUH1. Regarding the war period, our data does indicate a decrease in cropland which is consistent with literature reports, while LUH1 reports only a slight decrease, since the latter is extrapolated based on total population statistics, as discussed in p2 lines 25-30.

Furthermore, nowhere in the paper do we hypothesize that all occupied land was abandoned. In fact, the referee may verify the data presented in Supplement where we present cropland areas for occupied and non-occupied territories (reproduced below):

|                 |                             | 1940  | 1941  | 1942 | 1943 | 1944  | 1945  |
|-----------------|-----------------------------|-------|-------|------|------|-------|-------|
| Total           | FSU TOTAL                   | 150.4 | 108.1 | 87.7 | 94.1 | 109.9 | 113.6 |
| cropland        | OCCUPIED AREA 1940-1945     | 70.8  |       |      | 23.1 | 46.2  | 51.3  |
| (Million
ha) | NON-OCCUPIED AREA 1940-1945 | 72.7  | 74.9  | 77.7 | 66.4 | 59.0  | 57.6  |

These numbers were collected from: National economy of the USSR in the World War II (1941-1945). (Statistical Digest), Chapter 13: Agriculture (pp. 83-92). Goskomstat USSR, Information and Publishing Center, Moscow, 235 pp., 1990. (Narodnoe xozyajstvo SSSR v Velikoj Otechestvennoj vojne 1941-1945 gg. (Statisticheskij sbornik)., Goskomstat SSSR., Glava 13: Selskoe xozyajstvo (str. 83-92), Informacionno-Izdatelskij Centr, Moskva, 235 s., 1990), (in Russian).

Even though there is a gap of two years in 1941 and 1942, it is evident that the decrease in cropland area was not all located in the occupied territory.

RC3: Authors also did not consider the role of trade and food supply by the Soviet Allies (for instance, in the States we see cropland expansion over this period) and C emissions associated with deforestation during the war, fires and such and spatially differentiated land use displacement (it would be best at least to go with stats at the subnational level).

**AR3:** This is a good point for the global LUC emission budget (which we do not try to re-evaluate here). According to the US government census (figure below), cropland increased only moderately during this period, from ca. 320

MAcres in 1939 (129Mha) to 351 MAcres in 1944 (142Mha) and then 342 MAcres in 1949 (138Mha)\*. Thus, even though indeed there was an increase in cropland extent during the war period, it was rather moderate (13Mha), for such a large region.

---

## Editor Comment (EC1) · Y. Kuzyakov (Editor) · 5 Nov 2017

- The topic of the absence of CO2 increase in the 40ties is very relevant and still unclear. - The suggested 2 hypotheses: WWII and warm climate in this period are good and very interesting. - The Authors prepared a lot of materials reconstructing land cover and land use in the 40ties in the Former Soviet Union (FSU). There is not a lot of official (and probably not a lot of unofficial) materials about this time. Therefore, every additional data and opening of the data for broad communities are strong contribution to the retrospection and window to this difficult time. - The Authors made detailed analyses and modeling of the contributions of WWII and warming climate at this period

to the CO2 plateau.

I also looked on the comments of Anna Romanovskaya, anonymous Reviewer and the response of the Authors. As common, both reviewers mentioned some shortcomings and unclear points in the text and in the study. In my view, the Authors carefully responded to the Reviewers and probably will improve the paper for the next version.

Because of these reasons, I am sure that the paper should be published and surely will attract not only scientific but also general interests.

I should state – that I cannot evaluate the quality and the depth of modeling made by the Authors with ORCHIDEE, because this is not my expertise. However, 1) the obtained modeling results are very plausible comparing with C sequestration rates in soil and vegetation in abandoned land obtained experimentally in regions with similar climate and soils; and 2) Some of the Authors have excellent scientific records – so, I have no doubt that they have done and supervised the study on a very high level.

Suggestions (at least for Discussion) P3 L13 (here and in the main text too): 26.6 Mio are mentioned as killed during the WWII. This is not the only the reason for decreasing population density. I think the other very relevant reason should be mentioned: the victims of the repressions during the Stalin time (at least before WWII). To the directly killed $\sim$ 1 Mio people in the late 30ties, some millions (10 Mio-?) were departed from the west parts of the SU to the east and to the north and were spaced out from agriculture. This also led to the decrease of agricultural area and productivity. P6 L30 ... both scenarios are not clear. Explain with more details. P8 L7 What about the specially organized famine 1932/33? 6-7 Mio people (mainly in the west regions of SU) died in this period, and surely this led to the decrease of agricultural land area. This decrease is surely not reflected in the official land area statistics – because of various reasons, also because such area decrease is distributed very punctual and hardly to assess. I will not focus the paper on the political or social directions, but these huge losses of population (mainly in rural areas) surely led to consequences in subsequent

agriculture, including crop productivity and area.

There is another general point, which was not considered by the authors – It is the dynamics of C accumulation in vegetation and soil after the abandonment. In the first years (3-5-7 years, depending on the climatic zone) after the abandonment, the C stocks in soil decrease (not increase). This is connected with the time necessary for the establishing on natural zonal vegetation. (This was also indirectly mentioned by the anonymous Reviewer on page C4 in the middle). This establishment and succession takes much longer compared to the fast development of the annual sowed agricultural crops. Therefore, the compensation and overcompensation of soil organic matter losses as $CO_2$ by plant C input into the soil and stabilization takes years and decades. Therefore – and this is my opinion (and not the modeling), the reasons mentioned above related to Stalin repressions in the late 30ties and to the organized famine 1932/33 may have even stronger effects than the actual cropland area losses during WWII. For the warming hypothesis: P10 L8... The Authors compare the annual temperature and state that the 0.5 °C increase above the mean will lead to the strong $CO_2$ uptake by boreal and temperate forests in the latitudes above 45 °N. It is important that the annual temperature changes may not be really relevant, because e.g. the increase in the winter temperature of 1 °C will have not any effects on the $CO_2$ uptake. Therefore, the Authors should focus more (or at least present) the increase of the spring and summer temperatures, as well as on the prolongation of the vegetation period.

Yakov Kuzyakov 5.11.17

---

## Editor Comment (EC2) · Y. Kuzyakov (Editor) · 6 Nov 2017

Dear Authors,

here are additional comments from a reviewer. The reviewer is very critical to the quality and depth of your data collection and the modelling.

Sincerely yours, Yakov Kuzyakov

.......................................

Dear all,

[Figure]

Thanks for your heavy elaboration and clarification of the certain parts in the manuscript. This helped me a lot to understand your paper.

Let me shortly go over your major answers and point out the remaining critical aspects of your analysis and data you used, which I suggest improving and clarifying if you plan resubmit your paper to Biogesciences or to elsewhere. I have been noticed, your manuscript got already rejected for right now, but I still feel my second comments could help improving the manuscript.

1. You state, you are using state-of-the-art models and dataset to evaluate contribution of abandonment during WWII into CO2 plateau. I am more on the applied side of DVGM and other methods to estimate SOC dynamic due to LULCC, and this reflects, why I am so picky about utilized LULCC datasets. But let me add five cents on global DVGM and utilized datasets, such as HYDE 3.1. for LULCC modeling and carbon dynamic. Few words about HYDE. As you correctly stated in your manuscript, HYDE 3.1 FAOSAT data (cropland area) at the national level is implemented at country level. Cropland area does not track fodder crops, which is relevant for FSU. Thus, it does not match up data from Lyuri et al.,they used sown areas, which include fodder crops. It would be best to use here consistently the same source of information, such as stats from GOSKOMSTAT/ ROSSTAT, other official sources of FSU statistics on sown data. So, even if it is used widely for global studies (HYDE 3.1, K11), it does not necessarily mean the data is correct particularly for the regional studies, such as parts or entire FSU. For this reason, there were several studies, which tracked recently SOC due to LULCC for different parts of FSU, and then utilized, when it is possible, the official statistics at province level across different parts of FSU (Vuichard et al. 2008 http://doi.wiley.com/10.1029/2008GB003212, Kuemmerle et al. 2015 http://doi.wiley.com/10.1111/gcb.12897, Schierhorn et al. 2013 http://dx.doi.org/10.1002/2013GB004654).

Regarding, your modeling efforts, I did not find any contrasts with regional studies, which utilized DVGMs for your study area. Unfortunately, often validation of the outputs

of DVGMs is neglected produced numbers on SOC sequestration or release may vary quite a lot. As reviewer #1 also pointed out, you did not present and did not elaborate enough on validation of your model for such large region as FSU.

2. Now let me reflect on very important dataset you used to reconstruct cropland decline during WWII across FSU, namely reconstruction of sown area statistics. Thanks for clarification that you utilized Nove, Linz, Sapir, Davies for cross-reference and Maddison. I personally feel, you can delete this references and unnecessary text for few reasons. With the exception of few people, very few could easily access the stats from these books (I managed to download only work of Linz and just preordered the book of Nove). You could provide some permanent links to few key pages with data and key references to this data from Nove et al. I do not think, in this case, Nove publishing house would object. More importantly though, the sources of statistics, Nove and you utilized maybe the same, but not necessarily absolutely correct. There is no reason to argue that sown statistics from Goskomstat is better compared to FAOSTAT, so I would downplay spending so much time on data from other questionable sources (here I am asking why not then to use entirely stats on Gross Regional product and population from Soviet stats consistently?). But let me now shortly write down why statistics you utilized on abandoned land (declined crops) in your study requires additional elaboration or modification.

I managed to find and download the book you used. I am providing the link. https://drive.google.com/drive/folders/0Bzi0kSKsuOdgeFUzSFhraGFabWM?usp=sharing

You utilized the following statistics Goskomstat SSSR.

" These numbers were collected from: National economy of the USSR in the World War II (1941-1945). (Statistical Digest), Chapter 13: Agriculture (pp. 83-92). Goskomstat USSR, Information and Publishing Center, Moscow, 235 pp., 1990. (Narodnoe xozyajstvo SSSR v Velikoj Otechestvennoj vojne 1941-1945 gg. (Statisticheskij sbornik)., Goskomstat SSSR., Glava 13: Selskoe xozyajstvo (str. 83-92), Informacionno-

Izdatelskij Centr, Moskva, 235 s., 1990), (in Russian). Even though there is a gap of two years in 1941 and 1942, it is evident that the decrease in cropland area was not all located in the occupied territory."

If to look at the publication, this is minor, you provided wrong set of pages, this is surely minor (pp. 83-103). The most important though, the years for which there has been reported sown area statistics for the occupied area, namely 1940, 1943, 1944, 1945. By 1943 (and here we do not know if it was assessed for January 1st or by December 1943) there has been reported 23.1 Mha. However, by this time only a fraction of occupied area has been freed from Nazis. This means, Soviets could report only for those lands, which were able to control. I fully agree, a portion of abandoned lands on liberated territories could be abandoned (we also do not when abandonment actually started in 1941 or 1942). So only by 1945 the largest territory of FSU (including the Baltics) has been freed and here we confidentially by taking a difference between 1941 and 1945 to be on conservative safe side. Here we have then 19 Mha. It is much less than 64 Mha, but you can ensure avoiding issues with a lack of information for uncontrolled territories. Also a large portion of abandoned lands was actually abandoned for a year or two. This was also a reason, why I suggested to spatially differentiate and account in your models, where and when abandonment occurred. But I did not find any such spatial adjustment for your regional study.

I became further intrigued how Soviets could collect stats on occupied territories and decided to look at the definitions (p.4 National economy of the USSR in the World War II (1941-1945).). Here in Predislovie in Russian it says: ÂńĐš ŃĄĐśĐ¿ŃĂĐ¡ĐÿĐžĐţ Đ£ŃĂĐÿĐšĐţĐřĐţĐ¡ŃŃ ĐťĐřĐ¡Đ¡ŃŃĐţ ĐůĐř 1940 Đÿ 1943-1945 кк. Đ£Đ¿ ŃĂĐřĐźĐ¿ĐĐřĐij, Đ£Đ¿ĐťĐšĐţĂкѼĐÿĐijŃĄŃ Đ¿ĐžĐžŃČĐ£ĐřŃĘĐÿĐÿ. ЧŃĂĐÿ Ń■ŃĆĐ¿Đij ĐůĐř ĐžĐřĐůĐťŃŃĐź ĐÿĐů ĐšĐ¿ĐţĐ¡ŃŃŇ ĐžĐţŃĆ (1943-1945 кк.) Đ£ŃĂĐÿĐšĐţĐřĐţĐ¡ŃŃ ĐťĐřĐ¡Đ¡ŃŃĐţ Đ£Đ¿ Đ¿ĐśĐžĐřŃĄŃĆŃŔĐij Đÿ ŃĂĐřĐźĐ¿Đ¡ĐřĐij, ĐžĐ¿ŃĆĐ¿ŃĂŃŃĐţ Đš ŃĆĐţĐžŃČŇĽĐţĐij ĐÿĐžĐÿ Đ£ŃĂĐţĐťŃŃĐťŃĆŃĽĐÿŇ ĐşĐ¿ĐťĐřŇ ĐśŃŃĐžĐÿ

Ð¿ÑĄĐšĐ¿ĐśĐ¿ĐűĐřĐţĐ¡ÑŃ Đ¿ÑĆ Đ¿ĐžĐžŃČĐ£ĐřÑĘÿĐÿ. ĐŮĐř 1940 ĐşĐ¿Đř Đš Ñ■ÑĆĐÿÑĚ ÑĆŃМОпÑĘĐřÑĚ ĐśÑŃĐžÿ Đ£ÑĂĐřĐšĐřĐřĐ¡ÑŃ ĐřĐ¿ĐšĐ¿ĐţĐ¡ÑŃĆ ĐřĐřĐ¡ÑŃĆ Đ£Đ¿ ÑĆĐţÑĂÑĂĐÿÑĆĐ¿ÑĂĐÿĐÿ Đ¡Đř ĐijĐ¿ĐijĐţĐ¡ÑĆ ĐţĐ¡Đţ ĐijĐřĐžÑĂĐÿĐijĐřĐžÑŃĆĐ¿Đź Đ¿ĐžĐžŃČĐ£ĐřÑĘÿĐÿ Đš 1941- 1942 кк.)Âż. I am translating this. ÂńIn the compendium there have been provided numbers for 1940 and 1943-1945 for the areas(can be also interpreted-districts or rayons), which suffered from the occupation. At the same time, for each of the war years (1943-1945) there have been provided the numbers for oblasts and areas (can be also interpreted-districts or rayons), which in the current and preceeding years were liberated from the occupation. For 1940 in tables there have been provided prior the war data for the territory of the maximum occupation in 1941- 1942Âż. This confirms my assumption that stats you used is representative for freed territory by 1943 but not for entire occupied area, however, we do not know exactly if this was reported by jan 1, 1943 or December 31, 1943.

Again, to regionally fine tune your study, I recommend to account for the spatial location of the occupation and a fraction of the potentially occupied area. You may consider to reconstruct land use at province (oblast) level (this stats is available), and then make plausible projections on abandonment for occupied provinces for eahc specific year. Solely relying on country level data complemented with global data, is not sufficient to disentangle regional spatially differentiated processes.

Here are two examples of the maps on the advancement and retreat of Nazis

http://press.princeton.edu/chapters/haywood/s5_9519.pdf

https://en.wikipedia.org/wiki/Eastern_Front_(World_War_II)#/media/File:Eastern_Front_1943-08_to_1944-12.png

3. 13 Mha of cropland expansion for USA it is a large number as for any country too, including FSU. Taking into account large uncertainties with abandonment during WWII and more realistic 19Mha, this number has to be taken into account. I would

still heavily elaborate and contrast with other contributions to CO2 emissions, since AFOLU represents roughly 21 from total anthropogenic emissions of CO2. Reviewer 1, correctly pointed out, the importance to account for fires, burning, heavy extraction or forests by Nazis on occupied areas (e.g., Smolensk region). You take a hard task –to deal with large uncertainties with the numbers/ data you use and modeling approach, and you need to account for factors, which may balance out CO2 sink, in order to trust your numbers.

4. As you pointed out, and I reread your modeling approach, it would be best to avoid wording such as immediate forest regrowth, afforestation, rather establishment of seeds, shrubs regrowth. If you will spatially differentiate occupied lands, where occupation occurred, you will notice, a large portion of lands experienced SOC loss (thanks of for your figure). Even it has been occupied a large portion of temperate and northern regions, contribute probably not that much regarding cropland extent, compared to forest-steppe and southern regions (thanks for explanatory figure on spatially differentiated C pools and sinks).

Additional remarks.

L.15 p6. Howe sensitive your model to this threshold 0.85. how important this number compared to many other assumptions?

Table A1. Is it based on field data? How do these numbers vary across the study area? Some additional information on these numbers would be helpful.

Figure 1. I would just stick to your major storyline and will retain only FSU-Ref and FSE-New. Too much unnecessary details, with most likely, repeatable and questionable data.

To sum up, the storyline on wars and catastrophes and any socio-economic and environment shocks regarding land use and C dynamic is interesting and hot. However, the data you used (primarily data) and some assumptions right now downplay the validity of your story and claims. This certainly makes for right now feeling contribution of land abandonment to explain the plateau is dubious. However, surely, any large scale abandonment represents a certain C sink, but how other factors may counterbalance such sink, have to be accounted as well. Nevertheless, I pointed the options to improve the manuscript to address raised issues well and to make your findings stronger and more trustful.

---

## Author Comment (AC3) · 17 Nov 2017

We have fully revised the manuscript taking in consideration the issues raised by the previous editor, the two referees, the new review by referee #2, as well as the comments in EC3. In this regard, the main changes to the manuscript, apart from a thorough revision of the writing and section restructuring to improve readability and clarity are:

- Better description of the motivation for this study in the introduction, including the rationale behind the two hypotheses (climate vs land-use) proposed in Bastos et al. (2016). Particularly, we discuss in more detail the spatial and seasonal warming anomalies in

the 1940s and provide a justification for the choice of FSU as a study case;

- Improved the description of the LUC datasets, their characteristics and limitations, and better justification for collecting official statistics;

- Detailed explanation of the two scenarios and of the conversion from FSU totals to spatially-explicit maps, with reference to previous works using similar methods; - Justification for running the simulations up to the late 20th century (in order to compare our simulated Cstocks and fluxes with observation-based data);

- We now discuss the 1930's crisis. We would like to note that our collected data do not show a big decrease in cropland area during this period, for the reasons already mentioned by the Editor. Nevertheless, our data shows a small decrease in crop area between 1930 and 1933, which is not captured in LUH1. In the absence of alternative data, we decided to keep the values from the official statistics;

- We have dedicated special attention to the dynamics of soil C following abandonment and introduce now a Table with reference values of C-sequestration in former arable lands after the post-FSU abandonment in the 1990s. We would like to point out that all these studies point to a C-stock increase, rather than decrease, even in the first years following abandonment(1,2), at least in the Russian territory;

- We now show the spring and summer warming anomalies together with the annual values, and discuss how spring warming combined with mild summers might have contributed to such a strong enhancement of the terrestrial sink.

We are confident that these changes address the main issues raised regarding the motivation for this study, the relevance and reliability of the new LUC data collected, the ability of the model to simulate Russian C-stock dynamics following abandonment. We have chosen to focus the analysis on the available data and avoid introducing speculation about bombings and fire impacts on C-stocks. We provide a revised version of the manuscript with track changes highlighted. All authors have approved the revision

of the manuscript.

1 Kurganova, I.; De Gerenyu, V. L.; Shvidenko, A. & Sapozhnikov, P. Changes in the organic carbon pool of abandoned soils in Russia (1990–2004) Eurasian Soil Science, 2010, 43, 333-340.

2 Kurganova, I.; Lopes de Gerenyu, V.; Six, J. & Kuzyakov, Y. Carbon cost of collective farming collapse in Russia Global change biology, Wiley Online Library, 2014 , 20 , 938-947.

Please also note the supplement to this comment:
https://www.biogeosciences-discuss.net/bg-2017-267/bg-2017-267-AC3-supplement.pdf

---

## Author Comment (AC4) · 17 Nov 2017

**The contribution of land-use change versus climate variability to the 1940s CO2 plateau: Former Soviet Union as a test case**
**Bastos et al. *Biogeosciences**

**Response to Referee #2 (Editor's comment #2)**

*Thanks for your heavy elaboration and clarification of the certain parts in the manuscript. This helped me a lot to understand your paper. Let me shortly go over your major answers and point out the remaining critical aspects of your analysis and data you used, which I suggest improving and clarifying if you plan resubmit your paper to Biogesciences or to elsewhere. I have been noticed, your manuscript got already rejected for right now, but I still feel my second comments could help improving the manuscript.*

The manuscript has been proposed for reconsideration after major revisions, to be submitted by December 4, which we are carefully preparing, addressing the previous reviewers' comments. We thank the reviewer for taking the time to provide additional comments, even if we find it rather surprising to receive a second review, without having been given the opportunity to submit a revision of the manuscript. Nevertheless, in the revision we will submit, we have addressed the main points raised also here, to which we provide a point-by-point reply below.

*RC1: 1. You state, you are using state-of-the-art models and dataset to evaluate contribution of abandonment during WWII into CO2 plateau. I am more on the applied side of DVGM and other methods to estimate SOC dynamic due to LULCC, and this reflects, why I am so picky about utilized LULCC datasets. But let me add five cents on global DVGM and utilized datasets, such as HYDE 3.1. for LULCC modeling and carbon dynamic. Few words about HYDE. As you correctly stated in your manuscript, HYDE 3.1 FAOSAT data (cropland area) at the national level is implemented at country level. Cropland area does not track fodder crops, which is relevant for FSU. Thus, it does not match up data from Lyuri et al.,they used sown areas, which include fodder crops. It would be best to use here consistently the same source of information, such as stats from GOSKOMSTAT/ ROSSTAT, other official sources of FSU statistics on sown data.*

**AR:** As mentioned in the previous reply to (AR16), we do not rely on Lyuri et al. (2010) to produce our dataset, because Lyuri et al. (2010) does not provide any values. The book includes though one figure (2.28) showing the evolution of cropland in the FSU that allows estimating (visually) a decrease in cropland extent of ca. 25Mha. As mentioned before, our data was collected from the official national statistics mentioned by the reviewer and has been organized in a consistent way by aggregating the different crop type categories reported in GOSKOMSTAT reports. Indeed, as written in lines 6-7, page 6, we do mention that our data includes fodder crops:

> *"The total agricultural area is divided into regional values when available and includes winter and spring crops, industrial crops and sown area for fodder."*

At the same time, we compare our data with LUH/HYDE, which are (as any data) subject to uncertainty, but they are among the state-of-the-art datasets used in carbon and LUC modelling. Therefore, to the best of our knowledge, these are the most reliable and consistent sources of information about cropland evolution in fSU. We should note that FAO records are also based on the same national statistics (but they only compile them from 1961 onwards). In the revised version of the manuscript we improved the data description.

*RC2: So, even if it is used widely for global studies (HYDE 3.1, K11), it does not necessarily mean the data is correct particularly for the regional studies, such as parts or entire FSU. For this reason, there were several studies, which tracked recently SOC due to LULCC for different parts of FSU, and then utilized, when it is possible, the official statistics at province level across different parts of FSU (Vuichard et al. 2008 http://doi.wiley.com/10.1029/2008GB003212, Kuemmerle et al. 2015 http://doi.wiley.com/10.1111/gcb.12897, Schierhorn et al. 2013 http://dx.doi.org/10.1002/2013GB004654). Regarding, your modeling efforts, I did not find any contrasts with regional studies, which utilized DVGMs for your study area.*

**AR:** The two studies pointed by the referee are an excellent terms of comparison with our study, since they both used DGVMs to track changes in C-stocks following abandonment. Like in our work, both studies cited rely on national statistics to create a spatially-explicit dataset to force their models. In order to do so, they rely on prior information about geographical distribution of cropland, in Vuichard et al. using the maps from from Hurtt et al. (2006) ("*The shrinking total cropland area is distributed using the spatial land use pattern calculated by the global modeling study of Hurtt et al. [2006] which accounts for the marginality of land in the abandonment process.*"), and in Schierhorn et al, by combining land-cover and satellite maps, with statistics of sown area ("*This procedure is a combination of satellite-based global land cover data sets, namely, Global Land Cover 2000 (GLC2000) [Bartholomé and Belward, 2005], MODIS Land Cover [Friedl et al., 2002], and GlobCover [Bicheron et al., 2008], and subnational statistics on sown area [...]*"). In fact, our method is analogous to the method used in Vuichard et al., only an updated version of spatial land use pattern was used, i.e. based on Hurtt et al. (2011), instead of Hurtt et al., (2006). Therefore, we are

confident that our production of the spatially-explicit maps from the national statistics and our model simulations follows commonly accepted procedures in comparable land-use modelling studies. We will, nevertheless, introduce a more detailed description of the methods used to produce the land-cover map used to force ORCHIDEE-MICT simulations.

*RC3: Unfortunately, often validation of the outputs of DVGMs is neglected produced numbers on SOC sequestration or release may vary quite a lot. As reviewer #1 also pointed out, you did not present and did not elaborate enough on validation of your model for such large region as FSU.*

**AR:** We would like to point out that an earlier version of the ORCHIDEE model has been used by Vuichard et al. (2008) in their modelling exercise for C-stock changes in Russia following the collapse of the Soviet Union. We agree that it is worth introducing a comparison of our results and other estimates of C-stock changes following land abandonment. Therefore, we will add a Table comparing post-abandonment changes in C sequestration rates in the 1990s-2000s in the territory of the fSU provided by the review of Kurganova et al. (2014), which is reproduced below:

**Table 3** Estimations of total carbon sequestration in former arable lands of Russia

| Period | Area (M ha) | Approach | Total C sequestration (Tg C) | Average rate of C sequestration (Mg C ha$^{-1}$ yr$^{-1}$) | Reference |
|--------|-------------|----------|------------------------------|-----------------------------------------------------------|-----------|
| 1990–2011 | 45.5* | Soil-GIS | 870 (254)* | 0.92 (0.28) | Present study |
| 1990–2011 | 45.5* | Approximation | 861 (646)* | 0.96 (0.72) | Present study |
| 1990–2006 | 30.2 | Soil GIS | 648 (47) | 1.26 (0.09) | Kurganova et al., 2010; |
| 1990–2006 | 30.2 | Approximation | 585 (33) | 1.14 (0.06) | Kurganova et al., 2010; |
| 1990–2005 | 27.9 | RothC model | 248 (37) | 0.55 (0.08) | Romanovskaya, 2008; |
| 1991–2000 | 20.0 | Orchidee model | 64 | 0.47 | Vuichard et al., 2008; ** |
| 1990–2004 | 34.0 | Approximation | 660 | 1.29 | Larionova et al., 2003 |

We would further like to note that Table 2 of the manuscript does provide a comparison of modelled C-stocks and fluxes with observation based datasets for the late 20$^{th}$ century, which is intended to evaluate the results of ORCHIDEE-MICT simulations.

*RC4: 2. Now let me reflect on very important dataset you used to reconstruct cropland decline during WWII across FSU, namely reconstruction of sown area statistics. Thanks for clarification that you utilized Nove, Linz, Sapir, Davies for cross-reference and Maddison. I personally feel, you can delete this references and unnecessary text for few reasons. With the exception of few people, very few could easily access the stats from these books (I managed to download only work of Linz and just preordered the book of Nove). You could provide some permanent links to few key pages with data and key references to this data from Nove et al. I do not think, in this case, Nove publishing house would object. More importantly though, the sources of statistics, Nove and you utilized maybe the same, but not necessarily absolutely correct. There is no reason to argue that sown statistics from Goskomstat is better compared to FAOSTAT, so I would downplay spending so much time on data from other questionable sources (here I am asking why not then to use entirely stats on Gross Regional product and population from Soviet stats consistently?).*

**AR:** We have restructured the text in order to be clear that we do not use these data to construct our cropland area dataset. They are only now mainly used to provide a rationale about the reasons to focus on the FSU and to complement the discussion. Although we understand the reviewers concern with the fact that we use data from books, many other studies rely on statistics that are only available in books (e.g. Klein Goldewijk et al. (2011)). To the best of our knowledge, it is not common practice to provide reproductions of book pages as suggested by the reviewer, most likely because it is not compatible with copy rights. We would like to point out that many public and university libraries may have copies of the books for open consultation.

We do not find in the manuscript any claim that Goskomstat is better than FAOSTAT. This is actually impossible, because FAOSTAT is available since 1961 onwards (as mentioned in the manuscript) and we use data up to 1961. In fact, this is precisely why from 1961 onwards we keep LUH/HYDE values (themselves partly based on FAOSTAT). What we compare are the values prior to 1961 – which do not rely on FAO data, but on an extrapolation of total population for each country – with the data collected in this work.

**RC5: But let me now shortly write down why statistics you utilized on abandoned land (declined crops) in your study requires additional elaboration or modification. I managed to find and download the book you used. I am providing the link.**
**https://drive.google.com/drive/folders/0Bzi0kSKsuOdgeFUzSFhraGFabWM?usp=sharing You utilized the following statistics Goskomstat SSSR." These numbers were collected from: National economy of the USSR in the World War II (1941-1945). (Statistical Digest), Chapter 13: Agriculture (pp. 83-92). Goskomstat USSR, Information and Publishing Center, Moscow, 235 pp., 1990. (Narodnoe xozyajstvo SSSR v Velikoj**

**Otechestvennoj vojne 1941-1945 gg. (Statisticheskij sbornik)., Goskomstat SSSR., Glava 13: Selskoe xozyajstvo (str. 83-92), Informacionno-Izdatelskij Centr, Moskva, 235 s., 1990), (in Russian). Even though there is a gap of two years in 1941 and 1942, it is evident that the decrease in cropland area was not all located in the occupied territory." If to look at the publication, this is minor, you provided wrong set of pages, this is surely minor (pp. 83-103). The most important though, the years for which there has been reported sown area statistics for the occupied area, namely 1940, 1943, 1944, 1945. By 1943 (and here we do not know if it was assessed for January 1st or by December 1943) there has been reported 23.1 Mha. However, by this time only a fraction of occupied area has been freed from Nazis. This means, Soviets could report only for those lands, which were able to control. I fully agree, a portion of abandoned lands on liberated territories could be abandoned (we also do not when abandonment actually started in 1941 or 1942). So only by 1945 the largest territory of FSU (including the Baltics) has been freed and here we confidentially by taking a difference between 1941 and 1945 to be on conservative safe side. Here we have then 19 Mha. It is much less than 64 Mha, but you can ensure avoiding issues with a lack of information for uncontrolled territories. Also a large portion of abandoned lands was actually abandoned for a year or two. This was also a reason, why I suggested to spatially differentiate and account in your models, where and when abandonment occurred. But I did not find any such spatial adjustment for your regional study. I became further intrigued how Soviets could collect stats on occupied territories and decided to look at the definitions (p.4 National economy of the USSR in the World War II _ (1941-1945).). Here in Predislovie in Russian it says: ´nDš Ñ ´AĐ´sĐ¿ÑAĐ¡ĐÿĐžĐ¸Đ£Ñ´AĐÿКиĐt'Đ¸Đ¡ÑN Đt'Đ´rĐ¿Đ¿Ñ´Nи Đ ¸Đˇr 1940 Đÿ 1943-1945 Đ ¸Đ ¸.Đ£Đ¿ÑAĐ´rĐ´zĐ¿Đ¸Đ´ij,Đ£Đ¿Đt'КиtÑ´AĐ ¸Ñ´LĐÿĐijÑ AÑRĐ¿ĐžĐžÑCĐ£Đ´rÑ EĐÿĐÿ.ЧÑAĐÿÑÑCĐ¿ĐijĐ uСrОдrĐ´ uĐt'ÑNĐ´ zĐÿĐ uĐšĐ¿Đ¸Đ¸Đ¡ÑÑEĐ´zĐtÑ ´C (1943-1945 Đ ¸Đ ¸s.)Đ£Ñ ´AĐÿКиĐt'Đ¸Đ¡Ñ N Đt'Đ´rĐ¿Đ¸ÑNĐt´ ¸Đ£Đ¿Đ¸Đ´sĐzСrĐ AÑ´CÑRĐijĐÿÑ´AСr Đ´zĐ¿Đ¸Đ´rĐij,ĐžĐ¿ÑCĐ¿Ñ´AÑNĐt´ĐšÑÑCĐt´ĐžĐžÑÑĽL'Đt´Đi j Đ´ÿĐ´zĐÿ Đ Ñ´AĐt´ĐÑNĐt'NÑCĽL'ĐÿÑ´E Đ ¸Đ¿Đt'Đ´rÑ´E иÑNĐt´zĐÿ Đ¿Ñ AĐšĐ¿Đ¸Đ¸Đ´ uĐt'Đ¸Đ¡Ñ´N Đ¿Ñ´C Đ¿ĐžĐžÑ´CĐ£Đ´rÑ EĐÿĐÿ.Đ UСr 1940Đ ¸Đ¿Đt' Đš ÑCĐÿÑ´E ÑCĐt´rĐ´sĐĞzĐÿÑ EĐ´rÑ´E иÑNĐt´zĐÿ Đ£Ñ´AĐÿКиĐt'Đ¸Đ¡ÑNĐt'Đ¿ĐšĐ¿Đ¸Đ¸Đ¡ÑNĐt´ Đt'Đ´rĐ¿Đ¸ÑNĐt´ Đ£Đ¿ Ñ´CиtÑ ´AÑ´ AĐ´ÿÑ´CĐ¿Ñ´AĐÿĐÿ Đ¡Đ´rĐijĐ¿ĐijĐt´ĐÿÑC иtиt´ ĐijĐ´rĐžÑ´ AĐÿĐijĐt´rĐzÑ´NĐt´¿Đ¸´z Đ¿ĐžĐžÑ´CĐ£Đt´rÑ EĐÿĐÿ Đš1941- 1942 Đ ¸Đ ¸.)´z. I am translating this. ´nIn the compendium there have been provided numbers for 1940 and 1943-1945 for the areas(can be also interpreted-districts or rayons), which suffered from the occupation. At the same time, for each of the war years (1943-1945) there have been provided the numbers for oblasts and areas (can be also interpreted-districts or rayons), which in the current and preceeding years were liberated from the occupation. For 1940 in tables there have been provided prior the war data for the territory of the maximum occupation in 1941- 1942´z. This confirms my assumption that stats you used is representative for freed territory by 1943 but not for entire occupied area, however, we do not know exactly if this was reported by jan 1, 1943 or December 31, 1943 Again, to regionally fine tune your study, I recommend to account for the spatial location of the occupation and a fraction of the potentially occupied area. You may consider to reconstruct land use at province (oblast) level (this stats is available), and then make plausible projections on abandonment for occupied provinces for eahc specific year. Solely relying on country level data complemented with global data, is not sufficient to disentangle regional spatially differentied processes. Here are two examples of the maps on the advancement and retreat of Nazis [http://press.princeton.edu/chapters/haywood/s5_9519.pdf](http://press.princeton.edu/chapters/haywood/s5_9519.pdf) https://en.wikipedia.org/wiki/Eastern_Front_(World_War_II)#/media/File:Eastern_Front_1943-08_to_1944-12.png**

AR: We agree with the referee that there is uncertainty regarding the numbers referring to occupied area, which we will mention in the revised version of the manuscript. However, to the best of our knowledge the cropland area for the regions occupied between 1941-42 was not collected (or at least not reported) at the time. Any projection of hypothetical abandonment scenarios will, in our view, only contribute to increase uncertainty rather than reducing it.

Furthermore, the reviewer mentions that GOSKOMSTAT and the book cited provide cropland area at oblast level for this period. However, we were not able to find any reference to such data in GOSKOMSTAT. In the book mentioned by the referee, a list of oblasts is indeed provided, but no values of cropland per oblast are mentioned. Regarding the data for 1943, the data provided is for the full year, which usually is referred to growing season period in the case of FSU. In any case, for the point of this study, we would like to point out that FSU-REF (LUH/HYDE) also does not include any kind of information about occupied versus non-occupied regions.

Even though there are illustrative figures about the area roughly occupied, we could not find (we did search in the preliminary stages of this work) a geo-referenced map of the areas under occupation for different years of WWII. Even if we chose to present the results with the more simplistic rule of removing cropland proportionally to the fraction of crop in each pixel (as described now in more detail in the revised paper), this does not mean we did not carefully evaluate whether our resulting land-abandonment during the war period was consistent with the changes reported in the cited literature (namely the strongest reduction in occupied/front regions). We show the per-pixel

reduction in cropland fraction between 1940 and 1942 in the following figure, and compare with one of the figures suggested by the reviewer:

[Figure]

Figure: Geographical distribution of changes in pixel fraction of cropland (positive means cropland loss) between 1940 and 1942.

The figure shows that the strong cropland area decrease in our dataset occurs mainly in occupied regions, but especially in the regions corresponding to the war front. Since the resulting pattern was consistent with the literature cited and provided a better fit to region-level statistics (Figure A2), this dataset was found appropriate for the purpose of our study (again, keep in mind we want to compare with LUH/HYDE, which does not have any wartime LUC signal). We did, however, make some preliminary tests in which the reduction in crop area occurred mainly in the occupied/front region, as shown in the figure below.

[Figure]

Figure: Cropland reduction between 1940-1942

The simulation in which abandoned areas area replaced by grassland PFTs (using the same procedure as for the other simulations) resulted in an additional sink from ELUC of 0.08PgC/yr, instead of 0.07 PgC/yr in $S_{GRA}$. This relatively small difference does not hamper our conclusion: that land-abandonment during WWII could contribute only a small fraction of the gap sink required to explain the plateau, as compared to natural climate variability.

**RC6: 3. 13 Mha of cropland expansion for USA it is a large number as for any country too, including FSU. Taking into account large uncertainties with abandonment during WWII and more realistic 19Mha, this number has to be taken into account. I would still heavily elaborate and contrast with other contributions to CO2 emissions, since AFOLU represents roughly 21 from total anthropogenic emissions of CO2. Reviewer 1, correctly pointed out, the importance to account for fires, burning, heavy extraction or forests by Nazis on occupied areas (e.g., Smolensk region). You take a hard task –to deal with large uncertainties with the numbers/ data you use and modeling approach, and you need to account for factors, which may balance out CO2 sink, in order to trust your numbers.**

**AR:** We understand the reviewer's point about the fact that other regions might present different LUC trajectories that might offset/reinforce the responses we find in the FSU. However, the goal of this paper is not to discuss global patterns of LUC during the WWII, but to compare – using FSU as a test region – the potential contribution of the two

processes proposed by Bastos et al. (2016) to an enhanced terrestrial sink during the 1940s. The other contributions of $CO_2$ emissions (and sinks) at global scale have been thoroughly discussed in Bastos et al. (2016) and are therefore out of the scope of this paper. Still, we would like to note that in the 1940s, LUC contributed about as much as fossil fuel burning to anthropogenic $CO_2$ emissions, and with much larger uncertainty (about 1.5PgC/yr, please see figure 2 in Bastos et al. (2016)). The large uncertainty in LUC reconstructions is thus, not particular to our study, but a problem inherent to LUC data collection, terminology and assumptions (see e.g. Gasser and Ciais (2013) or Pongratz et al. (2014)). Our perspective is that it is best to base our analysis on the reliable and traceable data we can collect (as we did in this study), rather than adding further assumptions about village burning, bombing, or forest extraction by Nazis.

*RC7: 4.  As you pointed out, and I reread your modeling approach, it would be best to avoid wording such as immediate forest regrowth, afforestation, rather establishment of seeds, shrubs regrowth.  If you will spatially differentiate occupied lands, where occupation occurred, you will notice, a large portion of lands experienced SOC loss (thanks of for your figure).  Even it has been occupied a large portion of temperate and northern regions, contribute probably not that much regarding cropland extent, compared to forest-steppe and southern regions (thanks for explanatory figure on spatially differentiated C pools and sinks).*

**AR:** We have corrected the manuscript accordingly to explain how post-abandonment succession is modelled in ORCHIDEE-MICT. We now also include also a comparison with reference works on C-stock changes in Russia or FSU territory (see reply to RC3).

**Additional remarks.**

**RC8: L.15 p6. Howe sensitive your model to this threshold 0.85. how important this number compared to many other assumptions?**

**AR:** The threshold 0.85 represents the fraction of aboveground NPP that is harvested, and therefore is not transferred to the soil through litter. This value affects crop harvest productivity (together with many other parameters, e.g. Vcmax) and emissions when land-use change occurs. The resulting low C input from crops to soil may lead to overestimates of C soil gain following abandonment if this value is too high. Because the resulting C-stocks and LUC emissions result from a combination of several parameters, including this one, we compare our results with reference values of C-stocks and fluxes in the observational record (Table 2), with crop harvest estimates from economic records (Figure 3) and include in the revised version of the manuscript a Table comparing our estimates to C-sequestration

**Table A1. Is it based on field data? How do these numbers vary across the study area? Some additional information on these numbers would be helpful.**

**AR:** We believe the reviewer refers to the Vcmax values. As in most DGVM simulations, Vcmax values do not vary regionally and are PFT-dependent (see Krinner et al., 2005; Zhu et al., 2015). In our simulations, and analogous to Vuichard et al. (2008), we use somewhat modified parameters for crops and grasslands as compared to the standard values used in Zhu et al. (2015) and Guimberteau et al. (2017). This is the reason why, for transparency, we present the values used in Table A1.

**Figure 1. I would just stick to your major storyline and will retain only FSU-Ref and FSE-New.  Too much unnecessary details,  with most likely,  repeatable and questionable data.**

**AR:** We agree that Figure 1 has too much information and thus now only present FSU-REF and FSU-New together with the population statistics relevant for the discussion about LUH/HYDE characteristics.

**To sum up, the storyline on wars and catastrophes and any socio-economic and environment shocks regarding land use and C dynamic is interesting and hot.  However, the data you used (primarily data) and some assumptions right now downplay the validity of your story and claims. This certainly makes for right now feeling contribution of land abandonment to explain the plateau is dubious. However, surely, any large scale abandonment represents a certain C sink, but how other factors may counterbalance such sink, have to be accounted as well. Nevertheless, I pointed the options to improve the manuscript to address raised issues well and to make your findings stronger and more trustful.**

**AR:** We agree that the LUC impacts on C-sticks following major socio-economic events is a hot topic, but we would like to call attention for the other relevant aspect of this work, that relates to the impact of climate variability in high-latitudes in terrestrial $CO_2$ uptake. As Kurganova et al. (2014) thoroughly summarized for the post-1990s land abandonment in FSU, C-stock change estimates are likely to vary considerably depending on the methods used. Also, as many works show (already cited here), emissions from LUC are subject to multiple sources of uncertainty, being

one of the least constrained terms of the global carbon budget. The dataset we collected here is provided by the same sources used in FAOSTAT (and thus LUH/HYDE) after 1961. Our work was to collect and harmonize the data for FSU, to produce an annually-resolved, spatially-explicit, dataset of cropland area variations in FSU over the period prior to FAOSTAT, following standard methods used in works also cited by the reviewer. Here, we discuss how using regional LU data that incorporates information about drastic effects of socio-economic crises might result in significantly different ELUC estimates (up to 70% difference in the 1940s). We chose the FSU because its territory encompasses a very large terrestrial sink, a considerable fraction of soil C-stocks, and because it experienced at the same time the two processes that Bastos et al. (2016) hypothesised could contribute to the plateau – major social and economic changes during the early 20[th] century, and particularly in WWII, as well as high latitude warming during the late 1930s to late 1940s (Overland et al., 2004). Our intention with this study is not to have a final word about the LU changes in FSU during WWII, nor do we claim that our results indisputably show that LUC explains the plateau. Rather, our goal was to weight the possible contributions of both LU and climate variability to an increased terrestrial sink during the plateau period. Indeed, we find that large-scale high-latitude warming might be a better candidate to explain a potential enhancement of the terrestrial sink. We are confident that the dataset collected is relevant for the LU community and that our approach and model used are among the state-of-the-art methods used by the community and are therefore, scientifically valid.

**References**:

Gasser, T. & Ciais, P. A theoretical framework for the net land-to-atmosphere $CO_2$ flux and its implications in the definition of "emissions from land-use change". Earth System Dynamics, 2013, 4, 171-186.

Guimberteau, M.; Zhu, D.; Maignan, F.; Huang, Y.; Yue, C.; Dantec-Nédélec, S.; Ottlé, C.; Jornet-Puig, A.; Bastos, A.; Laurent, P.; Goll, D.; Bowring, S.; Chang, J.; Guenet, B.; Tifafi, M.; Peng, S.; Krinner, G.; Ducharne, A.; Wang, F.; Wang, T.; Wang, X.; Wang, Y.; Yin, Z.; Lauerwald, R.; Joetzjer, E.; Qiu, C.; Kim, H. & Ciais, P. ORCHIDEE-MICT (revision 4126), a land surface model for the high-latitudes: model description and validation. Geosci. Model Dev. Discuss., Copernicus Publications, 2017, 1-65.

Klein Goldewijk, K.; Beusen, A.; van Drecht, G. & de Vos, M. The HYDE 3.1 spatially explicit database of human-induced global land-use change over the past 12,000 years. Global Ecology and Biogeography, Blackwell Publishing Ltd, 2011 , 20 , 73-86.

Krinner, G.; Viovy, N.; de Noblet-Ducoudré, N.; Ogée, J.; Polcher, J.; Friedlingstein, P.; Ciais, P.; Sitch, S. & Prentice, I. C.A dynamic global vegetation model for studies of the coupled atmosphere-biosphere system Global Biogeochem. Cycles, 2005, 19 , GB1015.

Kurganova, I.; Lopes de Gerenyu, V.; Six, J. & Kuzyakov, Y. Carbon cost of collective farming collapse in Russia. Global change biology, 2014, 20, 938-947.

Overland, J. E.; Spillane, M. C.; Percival, D. B.; Wang, M. & Mofjeld, H. O. Seasonal and regional variation of pan-Arctic surface air temperature over the instrumental record. Journal of Climate, 2004, 17, 3263-3282.

Pongratz, J.; Reick, C. H.; Houghton, R. & House, J. Terminology as a key uncertainty in net land use and land cover change carbon flux estimates. Earth System Dynamics, 2014, 5, 177-195.

Zhu, D.; Peng, S.; Ciais, P.; Viovy, N.; Druel, A.; Kageyama, M.; Krinner, G.; Peylin, P.; Ottlé, C.; Piao, S. & others Improving the dynamics of Northern Hemisphere high-latitude vegetation in the ORCHIDEE ecosystem model. Geoscientific Model Development, 2015, 8 , 2263-2283.

---

## Author Comment (AC5) · 17 Nov 2017

**The contribution of land-use change versus climate variability to the 1940s CO2 plateau: Former Soviet Union as a test case**
**Bastos et al. *Biogeosciences***

**Response to Editor's comment #3**

Please find below a point-by-point reply to this comment, which we are not sure is from the Editor, from reviewer #1 or from another reviewer, as no information about this comment is included in the tracking system. We would like to note that we received on Oct 23 an indication that the manuscript needed major revisions, based on the previous referees' comments, which we are currently addressing. All of the issues raised here match the same concerns posed by the initial two reviews, which we are confident that our revision of the manuscript (to be submitted until December 4) settles.

*"The contribution of land-use change versus climate variability to the 1940s CO2 plateau: Former Soviet Union as a test case" In this study, authors attempted to explain why the stabilization of atmospheric CO2 concentration was observed during the 1940s. Earlier, Bastos et al. 2016 have showed that the global CO2 budget in terrestrial ecosystems during this period has a gap sink of 0.4-1.5 PgC yr-1. To explain this gap, authors made 2 hypotheses: (1) huge land-abandonment due to the socioeconomic and demographic disruptions during World War II that might lead to an additional Csequestration and (2) the warming observed in the high-latitudes for the same period, which might cause the enhancement of the natural C sink to vegetation. Unfortunately, I see some very serious problems, which do not allow to support the publication of this MS. I could agree that croplands abandonment took place in FSU during WWII. However, the period when current croplands were not used was short – not more 1-2 yrs. People which stayed on territory occupied by Nazi, had to produce food for them and themselves. Besides, withdrawn area was much less than it reports in the study – 62 Mha between 1940-1943. Here I support completely anonymous Rev # 2. Authors often used incorrected agricultural statistics.*

**AR:** These concerns are the same as the ones posed by Referee #1 and Referee #2, to which we already replied elsewhere. Nevertheless, we would like to note:

- Whether there was or not land abandonment in large areas of FSU during WWII is not a matter of opinion, but has been reported extensively in the literature referred in our manuscript. As in all report-based statistics, uncertainty exists concerning the exact values, but still different datasets can be compared for their impact on resulting CO2 fluxes. Indeed the data we collected may present some discrepancies in the nazi-occupied area, but the alternative dataset (LUH/HYDE) does not report any cropland abandonment during this period, which is arguably more inaccurate. In this regard, the referee could explain why they find that extrapolating cropland area based on total population prior to 1961 is less uncertain than our data based on official statistics.

- The high uncertainty in LUH/HYDE due to the method used prior to 1961 is precisely why we took the effort to collect statistics that are reliable (up to the degree to which most official statistics are reliable), and why we decided to base our analysis solely on the data we were able to collect and not on personal impression about the extent of abandoned area. The referee does not indicate any reference that can provide alternative figures and question our number. This is due to the indeed scarcity of the data for the early 20[th] century. As it so often happens in science, these numbers can be revised if alternative data becomes available.

- Still, we emphasize that the goal of this paper is to **compare two datasets** that, to the best of our knowledge, attempt to provide estimate of cropland variations in FSU during the 20[th] century using valid datasets and methods. We do not claim our data are free from errors or uncertainty, nor is LUH/HYDE (see Gasser and Ciais (2013) or Pongratz et al. (2014)). Our work provides a comparison of two equality valid datasets and discuss their impacts in resulting fluxes from land-use change emissions.

*It's also important to consider that the abandonment of high fertility soils can led to C-losses especially during first years after withdrawal (Romanovskaya, 2008; Lyuri et al., 2010, Kalinina et al., 2015). Such situation took place in Ukraine and south part of Russia, which were occupied by Nazi during WWII. So, I guess, that the amount of C sequestrated due to agricultural land abandonment are strongly overestimated, and even some additional soil C can be released as CO2.*

**AR:** Kurganova et al. (2014) have reviewed the changes in soil C and C sequestration following land-abandonment in the 1990s. They provide values from several works, all pointing to an increase in C-sequestration and soil C following abandonment due to the collapse of FSU (see Table below).

**Table 3** Estimations of total carbon sequestration in former arable lands of Russia

| Period | Area (M ha) | Approach | Total C sequestration (Tg C) | Average rate of C sequestration (Mg C ha$^{-1}$ yr$^{-1}$) | Reference |
|---|---|---|---|---|---|
| 1990–2011 | 45.5* | Soil-GIS | 870 (254)* | 0.92 (0.28) | Present study |
| 1990–2011 | 45.5* | Approximation | 861 (646)* | 0.96 (0.72) | Present study |
| 1990–2006 | 30.2 | Soil GIS | 648 (47) | 1.26 (0.09) | Kurganova et al., 2010; |
| 1990–2006 | 30.2 | Approximation | 585 (33) | 1.14 (0.06) | Kurganova et al., 2010; |
| 1990–2005 | 27.9 | RothC model | 248 (37) | 0.55 (0.08) | Romanovskaya, 2008; |
| 1991–2000 | 20.0 | Orchidee model | 64 | 0.47 | Vuichard et al., 2008; ** |
| 1990–2004 | 34.0 | Approximation | 660 | 1.29 | Larionova et al., 2003 |

Also, Kurganova et al. 2010 provide a synthesis of site-level measurements of changes in soil C following land-abandonment in the 1990s.

[Figure]

$$C\ ac = -53\ln(D) + 250$$
$$R^2 = 0.63$$

**Fig. 1.** Relationships between the carbon accumulation in the 0- to 20-cm-thick layer and the time of restoration of the arable soils (the number of pairs is 41).

As the referee may note, they find C-accumulation in the soils even a few years after abandonment, with the highest accumulation rates in the early years after abandonment. They conclude: "*On the whole, the carbon sequestration (20 million t of C/yr) because of the removal of croplands from agricultural use is considerable in the carbon budget of the plant ecosystems in Russia.*"

Nevertheless, we agree that it is important to provide comparisons of our results to observation-based and modelled results for similar processes and there include now one Table in the revised version of the manuscript comparing our estimates of C-sequestration following abandonment with the values compiled by Kurganova et al. (2014) from several studies. In their review, C-sequestration in former arable soils during the first 15 years following abandonment varies between 47-129 gC/m2/yr depending on the method. In our simulations, absolute C-sequestration in former arable soils during the first 15-yrs following the large cropland decrease in 1942 are 101-130 gC/m2/yr, for $S_{FOR}$ and $S_{GRA}$ respectively. Therefore, even if our values are in the higher range of the values provided in Kurganova et al. (2014), they are in line with previous studies.

*For correct estimation of C budget on FSU territory during the 1940s, authors have to take account of other disturbances which might to impact on C balance. Here are some of them: - large amount of CO2 emitted to atmosphere as a result of forest fires, burning biomass in abandoned fields, fires in thousands of villages and many big cities on the occupied territory; - decrease C sink due to deforestation in European and especially East part of FSU to provide the functioning of military factories and heating of housing possible only by wood during the WWII; - possible decrease of emitted CO2 due to collapse in transport, coal industry, and industry during first 1-3 years of WWII. Of course, it's very complicated aim to estimate all disturbances caused by WWII, but authors have to be more thoughtful and interpret more carefully statistical data and results of modelling.*

As mentioned in the manuscript, our model does account for forest fires (the referee may note that fire emissions are provided in Figure A4). As we mention in the text, the model incorporates the SPITFIRE module and has been shown to be able to simulate boreal fires (Yue et al., 2014; 2015). As for fires related to war in villages and fields or bombing, we do not have any data that could allow considering such processes, but lower population density is usually associated with less ignitions (Knorr et al., 2016). Likewise, these processes are not included in any of the estimates of ELUC using LUH/HYDE, and therefore, since we are **comparing two analogous datasets**, they should not affect the conclusions obtained through such comparison.

As for fossil fuel emissions, as discussed extensively in Bastos et al. (2016), uncertainty in fossil fuel emissions is very small (0.1PgC/yr) during this period, while for LUC emissions uncertainty is more than ten-fold higher (1.5PgC/yr). Again, we would like to remember that the goal of this paper is not to provide an exhaustive analysis of the global CO2 budget during WWII (redundant with Bastos et al. 2016), but to perform a comparative analysis of processes that could explain a terrestrial C sink higher than the one estimated by current state-of-the-art

reconstructions. As discussed in replies to the other referees and the editor our approach and model used are among the state-of-the-art methods used by the community and are therefore, **scientifically valid**.
We have re-structured the manuscript, included additional figures and comparisons with observation-based datasets, and therefore believe our revision of the manuscript (yet to be submitted) addresses all the referee's concerns.

**References**:

Gasser, T. & Ciais, P. A theoretical framework for the net land-to-atmosphere $CO_2$ flux and its implications in the definition of "emissions from land-use change". Earth System Dynamics, 2013, 4, 171-186.

Knorr, W.; Arneth, A. & Jiang, L. Demographic controls of future global fire risk Nature Climate Change, Nature Research, 2016 , 6 , 781-785

Kurganova, I.; De Gerenyu, V. L.; Shvidenko, A. & Sapozhnikov, P. Changes in the organic carbon pool of abandoned soils in Russia (1990--2004) Eurasian Soil Science, Springer, 2010 , 43 , 333-340

Kurganova, I.; Lopes de Gerenyu, V.; Six, J. & Kuzyakov, Y. Carbon cost of collective farming collapse in Russia. Global change biology, 2014, 20, 938-947.

Pongratz, J.; Reick, C. H.; Houghton, R. & House, J. Terminology as a key uncertainty in net land use and land cover change carbon flux estimates. Earth System Dynamics, 2014, 5, 177-195.

Yue, C.; Ciais, P.; Cadule, P.; Thonicke, K.; Archibald, S.; Poulter, B.; Hao, W.; Hantson, S.; Mouillot, F. & Friedlingstein, P. Modelling the role of fires in the terrestrial carbon balance by incorporating SPITFIRE into the global vegetation model ORCHIDEE--Part 1: simulating historical global burned area and fire regimes Geoscientific Model Development, Copernicus GmbH, 2014, 7, 2747-2767.

Yue, C.; Ciais, P.; Cadule, P.; Thonicke, K. & van Leeuwen, T. T. Modelling the role of fires in the terrestrial carbon balance by incorporating SPITFIRE into the global vegetation model ORCHIDEE --Part 2: Carbon emissions and the role of fires in the global carbon balance Geosci. Model Dev., Copernicus Publications, 2015, 8, 1321-1338.